# The impact of pyrethroid resistance on the efficacy and effectiveness of bednets for malaria control in Africa

Thomas S Churcher[1]*, Natalie Lissenden[2], Jamie T Griffin[1,3], Eve Worrall[2], Hilary Ranson[2]

[1]MRC Centre for Outbreak Analysis and Modelling, Infectious Disease Epidemiology, Imperial College London, London, United Kingdom; [2]Liverpool School of Tropical Medicine, Liverpool, United Kingdom; [3]Queen Mary's University, London, United Kingdom

**Abstract** Long lasting pyrethroid treated bednets are the most important tool for preventing malaria. Pyrethroid resistant Anopheline mosquitoes are now ubiquitous in Africa, though the public health impact remains unclear, impeding the deployment of more expensive nets. Meta-analyses of bioassay studies and experimental hut trials are used to characterise how pyrethroid resistance changes the efficacy of standard bednets, and those containing the synergist piperonyl butoxide (PBO), and assess its impact on malaria control. New bednets provide substantial personal protection until high levels of resistance, though protection may wane faster against more resistant mosquito populations as nets age. Transmission dynamics models indicate that even low levels of resistance would increase the incidence of malaria due to reduced mosquito mortality and lower overall community protection over the life-time of the net. Switching to PBO bednets could avert up to 0.5 clinical cases per person per year in some resistance scenarios.

**\*For correspondence:** thomas. churcher@imperial.ac.uk

## Introduction

It is estimated that 68% of the 663 million cases of malaria that have been prevented since the year 2000 have been through the use of long-lasting insecticide treated bednets (LLINs) (*Bhatt et al., 2015*). However, there is a growing realisation that insecticide resistance is putting these advances under threat (*WHO, 2012*), with mosquitoes reporting widespread resistance to pyrethroids, the only class of insecticides currently approved for use in bednets (*Ranson and Lissenden, 2016*). The public health impact of pyrethroid resistance in areas of LLIN use is hard to quantify as a comparison between sites is complicated by multiple epidemiological factors making it difficult to ascribe differences in malaria metrics solely to mosquito susceptibility (*Kleinschmidt et al., 2015*). The efficacy of LLINs against mosquitoes is typically measured in experimental hut trials (*WHO, 2013a*). These experiments are time consuming, relatively expensive, and geographically limited and by themselves they do not fully account for all effects of the LLIN as they do not show the community impact (herd effects) caused by the insecticide killing mosquitoes (*Killeen et al., 2007*; *Magesa et al., 1991*). Mathematical models can be used to translate entomological endpoint trial data into predictions of public health impact. Currently this has only been done for a small number of sites (*Briët et al., 2013*) making it difficult for malaria control programmes to understand the problems caused by insecticide resistance in their epidemiological setting.

There are no easy to use genetic markers that can reliably predict the susceptibility of mosquitoes to pyrethroid insecticide (*Weetman and Donnelly, 2015*). The current most practical phenotypic method for assessing resistance is the use of bioassays which take wild mosquitoes and measures

**eLife digest** In recent years, widespread use of insecticide-treated bednets has prevented hundreds of thousands cases of malaria in Africa. Insecticide-treated bednets protect people in two ways: they provide a physical barrier that prevents the insects from biting and the insecticide kills mosquitos that come into contact with the net while trying to bite. Unfortunately, some mosquitoes in Africa are evolving so that they can survive contact with the insecticide currently used on bednets.

How this emerging insecticide resistance is changing the number of malaria infections in Africa is not yet clear and it is difficult for scientists to study. To help mitigate the effects of insecticide resistance, scientists are testing new strategies to boost the effects of bednets, such as adding a second chemical that makes the insecticide on bednets more deadly to mosquitoes. In some places, adding this second chemical makes the nets more effective, but in others it does not. Moreover, these doubly treated, or "combination", nets are more expensive and so it can be hard for health officials to decide whether and where to use them.

Now, Churcher et al. have used computer modeling to help predict how insecticide resistance might change malaria infection rates and help determine when it makes sense to switch to the combination net. Insecticide-treated bednets provide good protection for individuals sleeping under them until relatively high levels of resistance are achieved, as measured using a simple test. As more resistant mosquitos survive encounters with the nets, the likelihood of being bitten before bed or while sleeping unprotected by a net increases. This is expected to increase malaria infections. As bednets age and are washed multiple times, they lose some of their insecticide and this problem becomes worse.

Churcher et al. also show that the combination bednets may provide some additional protection against resistant mosquitos and reduce the number of malaria infections in some cases. The experiments show a simple test could help health officials determine which type of net would be most beneficial. The experiments and the model Churcher et al. created also may help scientists studying how to prevent increased spread of malaria in communities where mosquitos are becoming resistant to insecticide-treated nets.

their mortality after exposure to a fixed dose of insecticide (**WHO, 2013a**). However the discriminating doses used in the assay are unrelated to the field exposure and so the predictive value of these bioassays for assessing the problems of pyrethroid resistance is unknown. A meta-analysis has shown that insecticide treated bednets still outperform untreated nets in experimental hut trials even against pyrethroid resistant populations (**Strode et al., 2014**) though the community impact (herd effects) of the LLIN was not assessed (**Killeen et al., 2007**). The population prevalence of pyrethroid resistance is known to be changing at a fast rate (**Toé et al., 2014**) making it important to regularly re-evaluate the efficacy of LLINs in order to guide current vector control and resistance management strategies (**WHO, 2012**).

There are limited tools available for tackling pyrethroid resistance and protecting the advances made in malaria control. Until new LLINs containing alternative insecticide are available the only alternative bednet are those containing pyrethroids plus the insecticide synergist piperonyl butoxide (PBO). Studies have shown that PBO LLINs are substantially better at killing insecticide resistant mosquitoes in some locations but not others (**Ngufor et al., 2014a, 2014b**; **Kitau et al., 2014**; **Asale et al., 2014**; **Ngufor et al., 2014c**; **Koudou et al., 2011**; **Corbel et al., 2010**; **Tungu et al., 2010**; **Malima et al., 2008**; **Adeogun et al., 2012a**; **Agossa et al., 2014**; **Malima et al., 2013**). PBO LLINs are more expensive than standard LLINs, with one manufacturer's 2012 price for PBO LLIN being US$4.90 compared to a comparable standard LLIN price of US$3.25 (**Briët et al., 2013**). This makes it unclear where and when their use would be beneficial over standard LLINs given constrained public health budgets. A mathematical modelling study used results from 6 experimental hut trials comparing a standard LLIN (PermaNet 2.0) with a PBO LLIN (PermaNet 3.0) against *Anopheles gambiae* sensu lato mosquitoes (**Briët et al., 2013**). It predicted that the more expensive PBO LLIN was still cost effective compared to a threshold of US$150/DALY averted (not comparing against standard LLINs) in 4 of the 6 sites, though these results are not generalisable beyond the

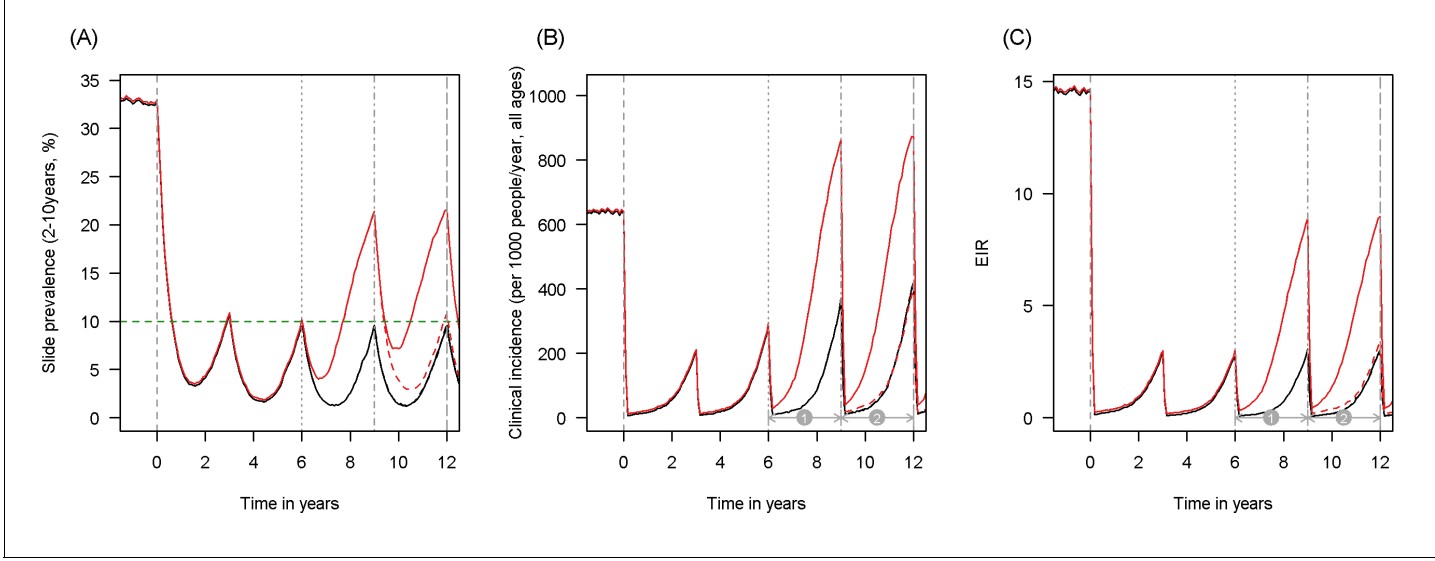

**Figure 1.** Scenario under investigation: timings for the introduction of LLINs, insecticide resistance and PBO LLINs for different malaria metrics. The figure illustrates how insecticide resistance is incorporated into the mathematical model. Panel (**A**) shows parasite prevalence by microscopy in 2–10 year olds, (**B**) clinical incidence in the entire population (cases per 1000 people per year) and (**C**) the annual entomological inoculation rate (EIR). In all three panels 4 different scenarios are run: black line shows a situation with no insecticide resistance whilst red line illustrates resistance arriving at year 6 (moderate, 50% survival measured in a bioassay); solid lines show non-PBO LLIN whilst dashed lines show PBO LLINs introduced at year 9 (vertical dotted-dashed grey line). There is no vector control in the population up until time zero (vertical dashed grey line) at which time there is a single mass distribution of non-PBO LLINs to 80% of the population. LLINs are redistributed every 3 years to the same proportion of the population. Mosquitoes are entirely susceptible up until resistance arrives overnight at the start of year 6 (vertical grey dotted line). Endemicity (a variable in *Figures 4* and *5*) is changed by varying the slide prevalence in 2–10 year olds at year 6 (by changing the vector to host ratio) and in this plot takes a value of 10% (as illustrated by the horizontal green dashed line in A). The impact of insecticide resistance is predicted (in *Figure 4*) by averaging the clinical incidence and EIR for the solid red lines (resistance) and solid black lines (no resistance) between the years 6 and 9 (period ①). Similarly, the impact of switching to PBO LLINs (in *Figure 5*) is estimated by averaging the clinical incidence and EIR for the solid red line (standard LLINs) and dashed red lines (switch to PBO LLINs) lines between the years 9 and 12 (period ②). Different scenarios with a low and high prevalence of pyrethroid resistance are shown in *Figure 1—figure supplements 1* and *2*.

The following figure supplements are available for figure 1:

**Figure supplement 1.** Scenario under investigation: example of a mosquito population with a low population prevalence of resistance.

**Figure supplement 2.** Scenario under investigation: example of a mosquito population with a high population prevalence of resistance.

specific sites chosen by the manufacturer, population prevalence of resistance, the type of LLIN or mosquito species. The WHO has recognised the increased bio-efficacy of PermaNet 3.0 in some areas (*WHO, 2015*) but there is a lack of clear consensus on when and where these should be deployed. Defining the added public health benefit expected by a switch to PBO LLINs is essential to guide decisions on pricing, purchasing and deployment.

Here we propose that information on the current malaria endemicity, mosquito species and population prevalence of pyrethroid resistance (as measured by bioassay mortality) can be used to predict the public health impact of pyrethroid resistance and choosing the most appropriate LLIN for the epidemiological setting. Firstly (1) a meta-analysis and statistical model are used to determine whether mosquito mortality in a bioassay can be used to predict the proportion of mosquitoes, which die in experimental hut trials and to define the shape of this relationship. Secondly (2), another meta-analysis of experimental hut trial data is analysed to characterise the full impact of pyrethroid resistance on LLIN effectiveness. Thirdly, information from (1) and (2) is used to parameterise a widely used malaria transmission dynamics mathematical model to estimate the public health impact of pyrethroid resistance in different settings taking into account the community impact of LLINs. An illustration of model predictions showing how different malaria metrics change over time is given in

**Table 1.** Summary of data collated in the three meta-analyses. The number of data points is subdivided according to the insecticides or LLIN tested and the predominant mosquito species in each population tested. Studies which did not determine species in the *Anopheles gambiae* complex are shown separately. All Published Data can be downloaded from Dryad Digital Repository whilst a list of the studies included their geographical range are given in the Material and methods.

| Meta-analysis description | Details | No. Studies | *Anopheles gambiae* s.s. | *Anopheles arabiensis* | *Anopheles gambiae* s.l. | *Anopheles funestus* | Total |
|---|---|---|---|---|---|---|---|
| *M1* Bioassay and experimental hut trial mortality | Deltamethrin | 5 | 2 | 1 | 10 | 0 | 13 |
| | Permethrin | 8 | 2 | 1 | 3 | 0 | 6 |
| | Other | 1 | 0 | 0 | 1 | 1 | 2 |
| | Total | 13 | 4 | 2 | 14 | 1 | **21** |
| *M2* Impact of PBO in pyrethroid bioassays | Deltamethrin | 16 | 15 | 5 | 29 | 8 | 57 |
| | Permethrin | 20 | 22 | 7 | 30 | 9 | 68 |
| | Other | 4 | 2 | 0 | 4 | 6 | 12 |
| | Total | 24 | 39 | 12 | 63 | 23 | **137** |
| *M3* Experimental hut trials of standard and PBO LLINS | Olyset | 6 | 6 | 0 | 10 | 0 | 16 |
| | PermaNet | 6 | 18 | 4 | 6 | 0 | 28 |
| | Total | 12 | 24 | 4 | 16 | 0 | **44** |

*Figure 1*. The figure also indicates how LLIN coverage and variables such as malaria endemicity are incorporated in the model. Finally (4) this model is combined with bioassay and experimental hut trial results to predict the epidemiological impact of switching from mass distribution of standard to PBO LLIN.

## Results

### Defining a metric for pyrethroid resistance

The population prevalence of pyrethroid resistance is defined from the percentage of mosquitoes surviving a pyrethroid bioassay performed according to standardised methodologies. Data from all bioassay types (such as the WHO tube susceptibility bioassay (*WHO, 2013b*), WHO cone bioassay (*WHO, 2013a*) or CDC tube assay [*Brogdon, 2010*]) are combined to produce a simple to use generalisable metric. Note that this pyrethroid resistance test does not differentiate between varying levels of resistance within an individual mosquito as only single discriminating doses are used. It is assumed that the ability of a mosquito to survive insecticide exposure is not associated with any other behavioural or physiological change in the mosquito population which influences malaria transmission. For example, an increased propensity for mosquitoes to feed outdoors (subsequently referred to as behavioural resistance) would limit their exposure to LLINs though there is currently insufficient field evidence to justify its inclusion in the model (*Briët and Chitnis, 2013*; *Gatton et al., 2013*).

### Using bioassays to predict LLIN efficacy

*Table 1* summarises the datasets used in the different meta-analyses. Meta-analysis *M1* shows that mosquito mortality in experimental hut trials can be predicted by the percentage of mosquitoes surviving a simple pyrethroid bioassay (*Figure 2A*). There is a substantial association between pyrethroid resistance in a bioassay and mortality measured in a standard LLIN experimental hut trial (*Figure 2A*, Deviance Information Criteria, DIC, with resistance as an explanatory variable = 2544.0, without = 2649.0 (lower value shows more parsimonious model), best fit parameters $\alpha_1$ = 0.634 (95% Credible Intervals, 95%CI, 0.012–1.29) and $\alpha_2$ = 3.99 [95%CI 3.171–5.12]). This indicates that bioassay survival can be used as a quantitative test to assess how the population prevalence of pyrethroid resistance influences LLIN efficacy. The number of studies identified in *M1* is relatively small

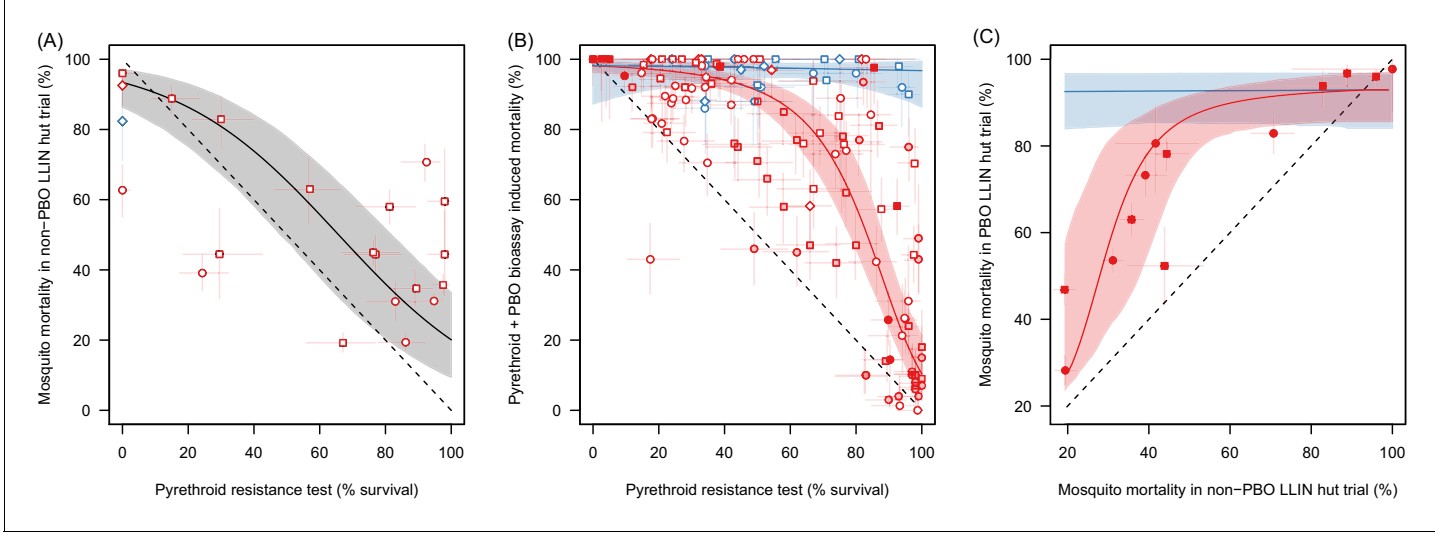

**Figure 2.** The ability of the pyrethroid resistance test (the percentage mosquito survival in a bioassay) to predict the results of experimental hut trials and the increase in mosquito mortality caused by the synergist PBO. Panel **A**: The relationship between mosquito mortality measured in non-PBO WHO tube bioassay and experimental hut trials (the percentage of mosquitoes, which enter the house that die within the next 24 hr). Solid grey line shows the best fit model for all mosquito species combined. Panel **B**: Differences in mosquito mortality caused by adding PBO to a pyrethroid bioassay. Panel **C**: Best fit models from Panel A and Panel B were combined to predict the change in mortality seen by adding PBO to a pyrethroid LLIN for mosquito populations with different levels of insecticide resistance. Points show the different mortalities measured from the limited number of experimental hut trials where PBO and non-PBO nets were simultaneously tested. Overall the model appears to be a good predictor of these data, both visually and statistically (Analysis of Variance test shows there was no significant difference between model predictions and observed data p-value=0.25). No experimental hut trial data were available for validation of the *Anopheles funestus* model. Throughout all panel colour denotes mosquito species, either *Anopheles gambiae sensu lato* (red) or *A. funestus* (blue), whilst the shape of points indicates the type of pyrethroid used: permethrin (circle), deltamethrin (square), or other pyrethroid (diamond). In panels **A** and **B** the fill of the points indicates the type of bioassay used (filled points = WHO cone; no fill = WHO tube; light fill = CDC bottle). Solid line shows the best fit model whilst the shaded areas indicate the 95% credible intervals around the best fit line. In all panels the dashed lines show no difference between the x and y axes. Pre-defined search string used in the meta-analyses are listed in *Figure 2—source data 1* whilst raw data from panels A,B and C are provided in *Figure 2—source data 2*, *Figure 2—source data 3*, and doi:10.5061/dryad.13qj2 respectively.

The following source data is available for figure 2:

**Source data 1.** Summary of the different predefined search strings used for meta-analysis *M1, M2 and M3*.

**Source data 2.** Summary of data from meta-analysis *M1* presented in Figure 2A.

**Source data 3.** Summary of data from meta-analysis *M2* presented in Figure 2B.

(only 21 data-points) so the predictive ability of the bioassay was further validated using the *A. gambiae s.l.*PBO data (*Figure 2B,C*).

## Added benefit of PBO

The increased mortality observed by adding the synergist PBO to a pyrethroid bioassay was assessed for *Anopheles funestus* and *Anopheles gambiae senu lato* mosquitoes with different levels of pyrethroid resistance (*M2*, *Figure 2B*). Data suggests that for the *A. gambiae* complex PBO has the greatest benefit in mosquito populations with intermediate levels of pyrethroid resistance (including pyrethroid resistance as an explanatory variable DIC = 2544.0, without DIC = 4748.0). In *A. funestus* adding PBO appears to kill all mosquitoes irrespective of the prevalence of pyrethroid resistance (including resistance as an explanatory variable improved model fit, with DIC = 2544.0, without DIC = 2547.0, though the gradient of the line was so shallow as to effectively make the PBO synergised pyrethroid mortality independent of the population prevalence of pyrethroid resistance).

The relationships identified in *Figure 2A and B* are used to predict the added benefit of a PBO LLIN over a standard LLIN (*Figure 2C*). These predictions are consistent with the observed results

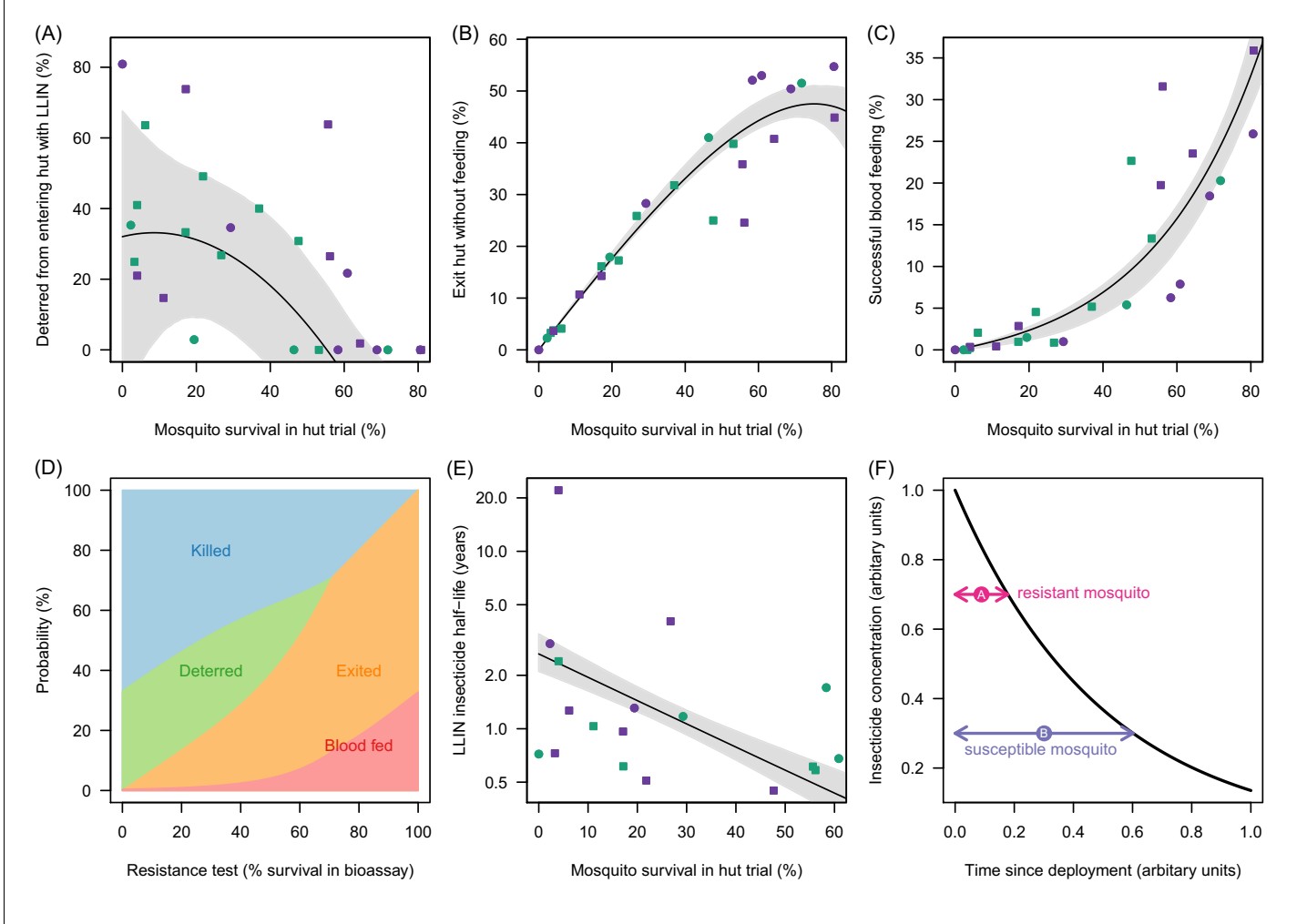

**Figure 3.** Meta-analysis of how the different outcomes of experimental hut trials which impact LLIN efficacy change with the percentage of mosquitoes which survive after entering the hut. (A) The probability that mosquitoes will be deterred away from a hut with an LLIN, (B) once entered the hut the mosquito will exit without feeding, or (C) will successfully feed. Panel (D) shows how the average probability that a bloodfeeding mosquitoes will be killed, deterred from entering, exit without feeding or successfully feed and survive during a single feeding attempt and how this changes with the population prevalence of pyrethroid resistance (as measured as the percentage survival in a pyrethroid bioassay). The lines are drawn using the best fit estimates from (A–C). Panel (E) shows how the longevity of the insecticide activity (estimated from washed nets) is longer in mosquito populations with high mosquito mortality in experimental hut trials. A possible hypothesis for this change is proposed in (F) where the black line indicates how insecticide concentration might decay over time. The time taken for a hypothetical resistant mosquito to survive the insecticide concentration (pink arrow) may be shorter than a susceptible mosquito (purple arrow). In Panels (A), (B), (C) and (E) the points show data from experimental hut trials with standard (green) or PBO (purple) LLINs. In (A) points which fell below the line (i.e. mosquitoes were more likely to enter huts with LLINs) were set to zero. The black line shows the best fit model to these data whilst the shaded area denotes the 95% credible interval estimates for the best fit line. Graphical assessment of the validity of the distributional assumptions and the posterior distributions for each parameter are shown in *Figure 3—figure supplement 1A*).

The following figure supplement is available for figure 3:

**Figure supplement 1.** Justification of normality distributed errors in the deterrence dataset (A) and posterior distributions of parameter estimates (B).

from all published experimental hut trials directly comparing both LLIN types (*M3*) (see overlap of data points with model predictions on *Figure 2C*) providing further independent evidence that the population prevalence of pyrethroid resistance measured by a bioassay can be used to predict LLIN induced mortality in a hut trial for both standard and PBO LLINs.

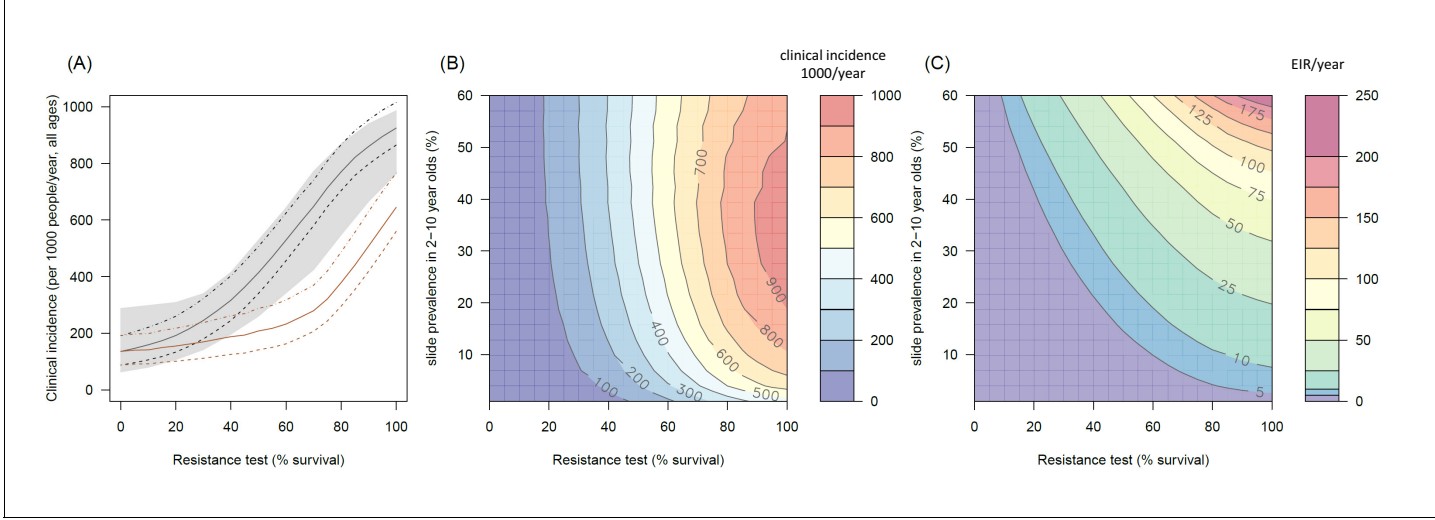

**Figure 4.** The predicted impact of pyrethroid resistance on the clinical incidence of malaria (Panels **A** and **B**) and the force of infection (Panel **C**). Panel (**A**) shows how the number of clinical cases in the population increases with the population prevalence of pyrethroid resistance (as assessed by the percentage survival in a pyrethroid bioassay) for a single setting (with 10% slide prevalence). Black lines show the full resistance model whilst the brown lines give predictions for mosquito populations where the rate of change in insecticide activity over time is the same for all mosquitoes (i.e. resistance has no impact on LLIN longevity). Solid lines show the average for the population, shaded grey area indicates the 95% credible intervals around this best fit line, dashed lines denote those using bednets whilst dotted-dashed lines show those who do not. Panel (**B**) shows the 3D relationship between prevalence of resistance (x-axis), endemicity (y-axis) and the absolute increase in the number of clinical cases (contours, see colour legend) per 1000 people (all ages). Panel (**C**) presents the same model as (**B**) though showing the absolute increase in the entomological inoculation rate (EIR, the average number of infectious bits per person per year). In this figure it is assumed that the mosquito species is *Anopheles gambiae sensu stricto* and that there is 80% LLIN coverage. *Figure 4—figure supplement 1* shows the same figure with 50% LLIN coverage. Further secondary figures indicate how the impact of resistance changes with mosquito species, be it *Anopheles arabiensis* (*Figure 4—figure supplement 2*) or *Anopheles funestus* (*Figure 4—figure supplement 3*). Panel (**A**) shows the importance of the rate of change in insecticide activity over time. *Figure 4—figure supplement 4* shows how Panels **B** and **C** would change if the rate of decay in insecticide activity was the same for resistant and susceptible mosquitoes. The uncertainty in the three LLIN efficacy parameters used to generate the confidence interval estimates in Panel (**A**) are shown in (*Figure 4—figure supplement 5*) for different levels of pyrethroid resistance.

The following figure supplements are available for figure 4:

**Figure supplement 1.** The predicted impact of pyrethroid resistance on the clinical incidence of malaria (Panels **A** and **B**) and the force of infection (Panel **C**) in an area with *A. gambiae s.s.* mosquitoes and 50% LLIN coverage.

**Figure supplement 2.** The predicted impact of pyrethroid resistance on the clinical incidence of malaria (Panels **A** and **B**) and the force ofinfection (Panel **C**) in an area with *A. arabiensis* mosquitoes and 80% LLIN coverage.

**Figure supplement 3.** The predicted impact of pyrethroid resistance on the clinical incidence of malaria (Panels **A** and **B**) and the force of infection (Panel **C**) in an area with *A. funestus* mosquitoes and 80% coverage.

**Figure supplement 4.** The predicted impact of pyrethroid resistance on (**A**) the clinical incidence of malaria and (**B**) the force of infection when pyrethroid resistance does not influence the rate of decay in LLIN insecticide activity over time (i.e. resistance has no impact on LLIN longevity).

**Figure supplement 5.** Estimates in the uncertainty of the three LLIN efficacy parameters for different levels of pyrethroid resistance.

## The impact of pyrethroid resistance on LLIN efficacy

Mortality in experimental huts was shown to be a useful predictor of LLIN induced deterrence, exiting and the rate of pyrethroid decay (*Figure 3A–C*). *Figure 3A* indicates that the number of mosquitoes deterred from entering the experimental hut substantially decreases in areas of higher pyrethroid resistance (where LLIN induced mortality inside the hut is low) though the variability around the best fit line is high suggesting the precise shape of the relationship is uncertain. As the population prevalence of pyrethroid resistance increases (and mortality inside the hut decreases) an

increasing proportion of mosquitoes entering the house exit without blood-feeding (*Figure 3B*). Only when there is a very high population prevalence of pyrethroid resistance does the probability that a mosquito will successfully feed start to increase (*Figure 3C*). Changing behaviour of a host seeking mosquito with different levels of pyrethroid resistance is shown in *Figure 3D*.

The overall efficacy of an LLIN depends on its initial efficacy and the rate at which this changes over the life-time of the net. Since there are currently no published durability studies in areas of high pyrethroid resistance or with PBO LLINs we estimate the loss of insecticidal activity from experimental hut trials using washed nets. Results indicate that washing decreases efficacy fastest in areas of higher pyrethroid resistance. *Figure 3E* shows estimates of the decay in pyrethroid activity assuming that the loss of efficacy due to washing is proportional to the change in activity seen over time (i.e. if the rate of decay over subsequent washes is twice as fast in a resistant mosquito population than the decay of pyrethroid activity over time will also be twice as fast). Mosquitoes with high pyrethroid resistance appear to overcome the insecticide activity of the LLIN faster than susceptible mosquitoes. A hypothesis for the cause of this relationship is outlined in *Figure 3F*.

## The public health impact of pyrethroid resistance

The transmission dynamics model predicts that the higher the population prevalence of pyrethroid resistance the greater impact it will have on both the number of clinical cases (*Figure 4A and B*) and the force of infection (as measured by the EIR, *Figure 4C*). This is due to the lower initial killing efficacy of the LLIN but also because of the higher rate of decay of insecticidal activity (it gets less effective more quickly). The absolute increase in EIR caused by resistance increases in areas of high endemicity (*Figure 4C*), though the model predicts that the number of clinical cases caused will peak at intermediate parasite prevalence because high levels of clinical immunity will mask increased infection rates in hyper-endemic areas. Understandably the impact of resistance will depend on the current LLIN coverage, with the total public health impact of resistance being greatest in areas where bednets were having the highest impact (i.e. areas of lower, 50%, coverage, see *Figure 4—figure supplement 1*). Equally the impact of resistance will be higher in areas with mosquito species which are more amenable to control through the use of LLINs (i.e. greater in *Anopheles gambiae sensu stricto* than *Anopheles arabiensis*, *Figure 4—figure supplements 2* and *3*). The transmission dynamics model predicts that the public health impact of pyrethroid resistance will be high. For example with as little as 30% resistance (70% mortality in discriminating dose assay) in a population with 10% slide prevalence (in 2–10 year olds) the model predicts that pyrethroid resistance would cause an additional 245 (95%CI 142–340) cases per 1000 people per year (*Figure 4A*, averaged over the 3 year life-expectancy of the net). Similar increases in the number of cases are seen in those with or without LLINs (*Figure 4A*).

## The public health benefit of switching to PBO LLINs

The impact of the addition of the synergist, PBO, on pyrethroid induced mortality appears to depend on mosquito species and the population prevalence of pyrethroid resistance. In mosquito populations with moderate to high resistance results indicate PBO is an effective synergist of pyrethroids (*Figure 5A*). For example in an area with 10% endemicity and 80% resistance (20% mortality in discriminating dose assay) the model predicts that switching to PBO LLINs would avert an additional 501 (95%CI 319–621) cases per 1000 people per year (*Figure 5A*) compared to the same level of standard LLIN coverage. The absolute number of cases averted by switching to PBO LLINs is predicted to be greater in areas with intermediate endemicity as human immunity is likely to partially buffer the added benefit of PBO LLINs in areas of highest malaria prevalence (*Figure 5B*). However, due to the non-linear relationship between incidence of clinical infection and endemicity, the greatest percentage reduction in clinical cases and EIR is seen in areas of low endemicity (Figure 5CF). The exact change in clinical cases will vary between settings. For example switching from 80% coverage with standard LLINs to 80% coverage with PBO LLINs in an area with 30% endemicity and a mosquito population with 60% pyrethroid resistance is predicted to reduce the number of clinical cases by ~60% whereas the same switch in the type of nets used in an area with 30% endemicity and 20% pyrethroid resistance would only reduce the number of clinical cases by ~20% (*Figure 5C*). Greater percentage reductions are likely to be seen in EIR than the number of clinical cases due to human immunity (*Figure 5E*).

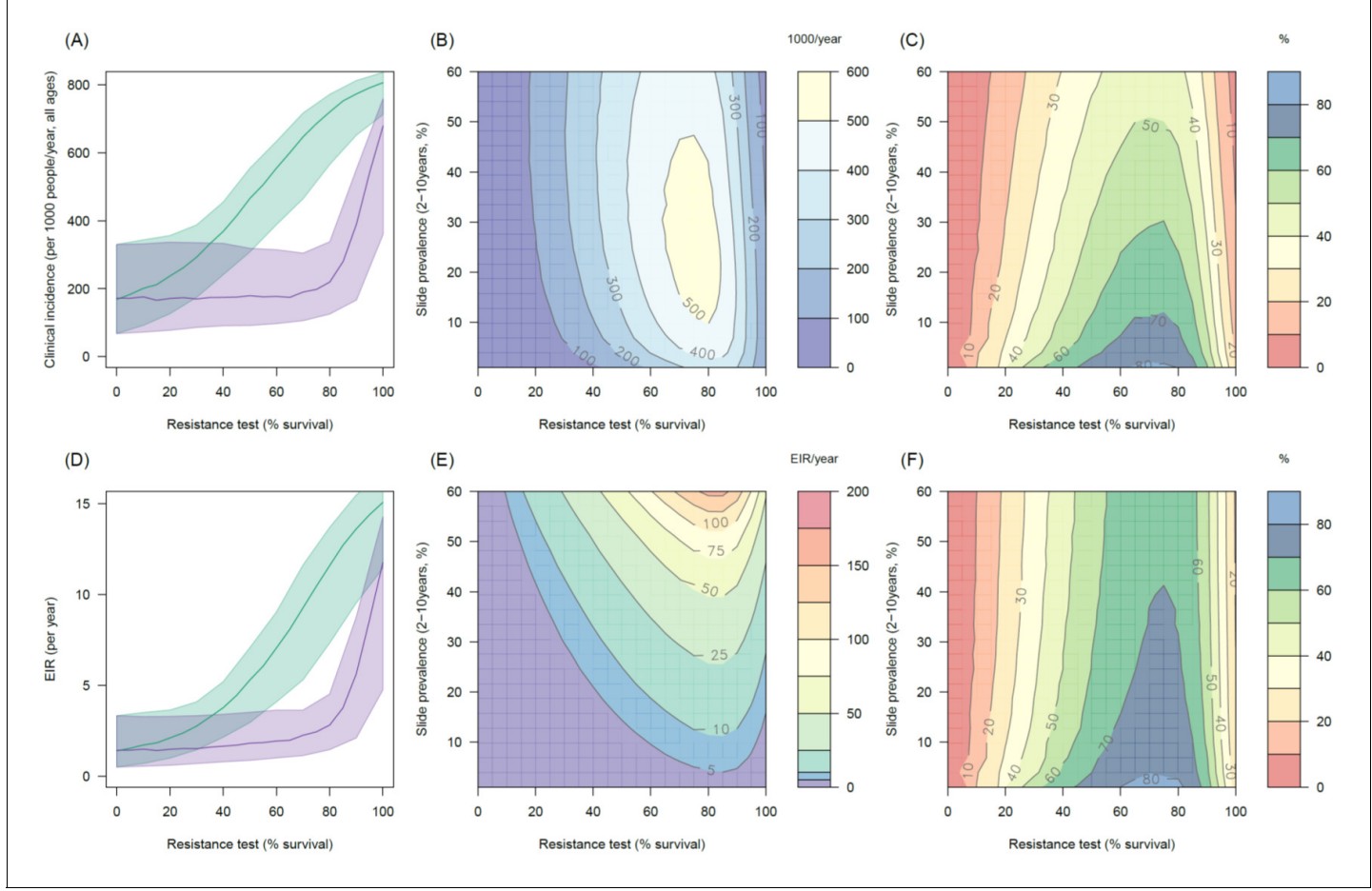

**Figure 5.** Predicting the added benefit of switching from standard LLINs to combination PBO nets. Panels (**A–C**) show clinical incidence (per 1000 people per year, all ages) whilst Panels (**D–F**) gives the entomological inoculation rate (EIR, infectious bites received per person per year). (**A**) and (**D**) show how malaria incidence and the force of infection increase with the population prevalence of pyrethroid resistance (as assessed by the percentage survival in a pyrethroid bioassay) in a single setting (with 10% slide prevalence) for standard LLINs (green line) and PBO LLINs (purple line). Shaded region denotes the 95% credible intervals around the best fit lines. Panels (**B**) and (**E**) show the 3D relationship between the prevalence of resistance (x-axis), endemicity (y-axis) and the absolute number of cases (and EIR) averted by switching to PBO LLINs. (**C**) and (**F**) give 3D relationship for the percentage reduction in cases and EIR (respectively) caused by switching from standard to PBO LLINs. The non-linear relationship between endemicity, clinical incidence and EIR means that the greatest percentage reduction is seen at low endemicities despite the greatest absolute reduction being in higher transmission settings. In all Panels it is assumed that the mosquito species is *Anopheles gambiae sensu stricto* and that there is 80% LLIN coverage. *Figure 5—figure supplement 1* shows the same figure with 50% LLIN coverage. Further secondary figures indicate how the impact of resistance changes with mosquito species, be it *Anopheles arabiensis* (*Figure 5—figure supplement 2*) or *Anopheles funestus* (*Figure 5—figure supplement 3*).

The following figure supplements are available for figure 5:

**Figure supplement 1.** Predicting the added benefit of switching from standard LLINs to combination PBO nets in an area with *A. gambiae s.s.* mosquitoes and 50% LLIN coverage.

**Figure supplement 2.** Predicting the added benefit of switching from standard LLINs to combination PBO nets in an area with *A. arabiensis* mosquitoes and 80% LLIN coverage.

**Figure supplement 3.** Predicting the added benefit of switching from standard LLINs to combination PBO nets in an area with *A. funestus* mosquitoes and 80% LLIN coverage.

# Discussion

Pyrethroid resistance is widespread across Africa though its public health impact is unknown. Here we show that the simple bioassay can be used to predict how pyrethroid resistance is changing the efficacy of different types of LLIN and how this would be expected to influence malaria morbidity.

The bioassay is a crude tool for measuring pyrethroid resistance, though its simplicity makes it feasible to use on a programmatic level. *Figure 2A and C* indicate that on average bioassay mortality is able to predict the results of standard and PBO LLIN experimental hut trials for *A. gambiae s.l.* mosquitoes. There is a high level of measurement error in the bioassay (as seen by the wide variability in points in *Figure 2A and B*) so care should be taken when interpreting the results of single assays as differences in mosquito mortality may have been caused by chance. Multiple bioassays could be conducted on the same mosquito population and the results averaged to increase confidence. However the exact cause of the measurement error remains unknown so increased repetition many not necessarily generate substantially more accurate results as possible causes of variability, such as mosquito husbandry techniques or environmental conditions (*Kleinschmidt et al., 2015*), may be repeated. Further work is therefore needed to determine whether assay repetition substantially improves overall accuracy or whether further standardisation or more complex assays are required. The majority of data are for *A. gambiae s.l.* so the analysis needs to be repeated for other species once data becomes available. More advanced methods of measuring insecticide resistance (such as the intensity bioassay [*Bagi et al., 2015*] or the use of genetic markers [*Weetman and Donnelly, 2015*]) are likely to be a more precise way of predicting resistance. However, since there are insufficient data to repeat these analyses with these other assays their predictive ability remains untested. Similarly, this analysis has grouped WHO tube, WHO cone and CDC bottle assays together when the use of a single assay type might be more predictive.

The meta-analysis of experimental hut trials in areas with different levels of resistance has important implications for our understanding of how pyrethroid resistance influences LLIN efficacy. This analysis suggests that the probability that a mosquito will feed on someone beneath an LLIN only increases substantially at high levels of pyrethroid resistance (*Figure 3C*). People under bednets exposed to mosquito populations with intermediate levels of resistance still have a high degree of personal protection whilst in bed as those mosquitoes, which do not die are likely to exit the hut without feeding. It is only when mosquito populations are highly resistant (>60% survival) that an increasing proportion of mosquitoes appear to successfully feed through the LLIN (*Figure 3D*). This may explain why a previous meta-analysis on the impact of pyrethroid resistance on LLIN efficacy in experimental hut trials failed to find a substantial effect (*Strode et al., 2014*) as resistance was categorised into broad groups (partially based on highly variable bioassay data) unlike here where resistance is treated as a continuous variable (as measured using experimental hut trial mortality data which are less variable than bioassay data). This earlier study also only analysed papers published or presented prior to May 2013 and so it did not include the recent experimental hut trials which had the lowest mosquito mortality (*Toé, 2015*; *Pennetier et al., 2013*).

The meta-analysis revealed that the number of mosquitoes deterred from entering a hut with an LLIN, decreases with increasing pyrethroid resistance. LLIN efficacy is therefore reduced as mosquitos enter huts where they have both a higher chance of feeding and a lower chance of being killed. These parallel changes in behaviour increase the resilience of mosquito populations to LLINs as in a susceptible mosquito population, high deterrence will reduce LLIN efficacy by preventing mosquitoes entering houses where they have a high chance of being killed (relative to susceptible populations). Importantly the loss of deterrence suggests that those sleeping in a house with an LLIN though not sleeping under the net themselves (a phenomenon particularly common in older children [*Nankabirwa et al., 2014*]) will lose an additional degree of protection (on top of the community impact of mosquito killing).

The overall effectiveness of LLINs depends on the duration of insecticidal activity. Evidence suggests that multiply washed LLINs lose their ability to kill mosquitoes more in areas of high pyrethroid resistance. Washing is seen as an effective method of aging LLINs (*WHO, 2013a*). Repeatedly washing a net (and presumably reducing the concentration of the insecticide) appears to have little impact on its ability to kill a susceptible mosquito whilst significantly reducing the lethality of the LLIN against more resistant mosquitoes (*Figure 2E*). The difference in mortality is likely to be caused by mosquitoes with a higher population prevalence of resistance being able to tolerate a higher

concentration of insecticide (*WHO, 2013a*). If so, then the higher longevity of LLINs against susceptible mosquitoes observed in the washed net data may be explained by the longer time it takes for the insecticide concentration on the LLIN to drop below this critical level (*Figure 2F*). This analysis assumes that the decay in pyrethroid activity over time is proportional to its decay following washing and this needs to be confirmed by durability studies in areas of high pyrethroid resistance. Nevertheless the results seem to be confirmed by two recent studies which evaluated mosquito mortality in older (standard) LLINs (*Toé et al., 2014*; *Wanjala et al., 2015*). Durability studies should be prioritised as the model predicts that, even at low levels of pyrethroid resistance, the loss of insecticide activity over the three year bednet life-expectancy, has a bigger epidemiological impact on malaria, than the initial efficacy of new LLINs. If confirmed then more regular net distribution could be considered as a temporary, albeit expensive, method to mitigate the public health impact of high pyrethroid resistance.

Transmission dynamics mathematical models are a useful tool for disentangling the different impacts of LLINs. Though a person under an LLIN requires high pyrethroid resistance before LLINs start to fail (*Figure 3C*), the models predict that at a population level even low pyrethroid resistance can increase the number of malaria cases over the life-time of the net (*Figure 4A*). Hut trials measure feeding when the volunteer is underneath a bednet whilst in reality (and in the mathematical model) a percentage of mosquito bites are taken when people are not in bed. The loss of LLIN induced mosquito mortality is likely to decrease the community impact of LLINs, increasing average mosquito age and the likelihood that people are infected whilst unprotected by a bednet. This is primarily due to the shorter duration of insecticide potency of LLINs in mosquito populations with a higher prevalence of resistance (*Wanjala et al., 2015*). Without this change in the duration of pyrethroid activity, the epidemiological impact of pyrethroid resistance will only become evident once it reaches a high level (*Figure 4A*). The change in the community impact of LLINs can be seen in the increase in the number of cases in people who do not use nets. This change is substantial, reinforcing the need to consider community effects in any policy decision.

Detecting an epidemiological impact of a low population prevalence of resistance may be challenging for local health systems (for example, see < 20% resistance prevalence *Figure 1—figure supplement 1*, *Figure 4*) especially in an area where LLIN coverage, local climatic conditions and the use of other malaria control interventions are changing over time. These simulations also assume that resistance arrives overnight, when in reality it will spread through a mosquito population more gradually and therefore may be harder to detect. Mosquitoes exposed to LLINs may have reduced fitness (*Viana et al., 2016*). Currently the model assumes that mosquitoes which survive 24 hr after LLIN exposure are indistinguishable from unexposed mosquitoes. If this is not the case then hut trials data alone will be insufficient to predict the public health impact of pyrethroid resistance as current models will over-estimate its impact. Similarly, if the mosquito population exhibits additional behavioural mechanisms to avoid LLINs, such as earlier biting times, in tandem to the increased tolerance of pyrethroid insecticide then the predictions presented here will likely underestimate the public health impact as this behaviour change has not been incorporated.

Currently a mosquito population is defined as being pyrethroid resistant if there is < 90% bioassay mortality (*WHO, 2013b*; *Mnzava et al., 2015*). Though useful, this entomological measure should not be considered as a measure of the effectiveness of pyrethroid LLINs. The personal protection provided by sleeping under an LLIN is likely to be substantial even at very high levels of resistance (*Strode et al., 2014*; *Randriamaherijaona et al., 2015*). Any reduction in mosquito mortality will likely reduce the community impact of LLINs though it may be hard to detect, especially in areas with new LLINs (the public health impact of resistance is likely to be greater in older nets, *Figure 3E*). As with all transmission dynamic mathematical models, these predictions need to be validated in particular locations with well-designed studies combining epidemiological and entomological data. We are currently unaware of any published data with sufficient information to test the model against though a thorough validation exercise should be carried out as soon as such studies become available. Currently the meta-analyses and transmission dynamics models concentrated on malaria in Africa and give predictions for the three primary mosquito vector species found there. Each meta-analyses has data from multiple countries but these sites are not geographically representative of the whole of malaria endemic Africa. Though the principles outlined here may apply to other mosquito species in different care settings should be taken when extrapolating the results beyond the areas where the data were collated.

The bioassay data indicate that the ability of PBO to synergise pyrethroid induced mortality depends on the mosquito species. In *A. funestus* PBO always appears to restore near 100% mortality whilst for mosquitoes from the *A. gambiae* complex the greatest additional benefit of PBO being seen at intermediate levels of pyrethroid resistance (*Figure 2B*). The exact causes of this are unknown but is likely related to the predominant resistance mechanisms in each species. PBO's primary synergistic effect of pyrethroids is thought to be due to the inhibition of the cytochrome P450 enzymes which catalyse the detoxification of the insecticides (*Farnahm, 1998*). Elevated P450 levels are the primary resistance mechanism in *A. funestus* whereas in *A. gambiae s.l.* both increased detoxification and alterations in the target site contribute to pyrethroid resistance with the latter mechanism being largely unaffected by PBO (*Mulamba et al., 2014*; *Riveron et al., 2013*).

For *A. gambiae s.l.* populations this result was verified by experimental hut trial data which directly compare standard and PBO LLINs (*Figure 2C*). Both bioassay and hut trial data suggest a minimal additional benefit of PBO in areas with very high levels of pyrethroid resistance. Unfortunately, there are currently no published studies where PBO LLINs have been tested in experimental hut trials in areas with *A. funestus* so these bioassay results should be treated with caution until they can be further verified. Additional data would also allow the differences between species in the *A. gambiae* complex to be assessed. A previous analysis comparing PermaNet 2.0 and 3.0 was unable to test whether the increase in efficacy of the PBO LLIN was solely due to the addition of PBO as this net has a higher concentration of insecticide (*Briët et al., 2013*). The results presented here show a consistent pattern between PermaNet 2.0 and 3.0 and Olyset and Olyset Plus. As both Olyset nets have the same concentration of insecticide, this suggests that PBO is causing the enhancement of efficacy.

The WHO recommends that countries routinely conduct non-PBO pyrethroid bioassays as part of their insecticide resistance management plan (*WHO, 2012*). In areas with *A. gambiae s.l.* the evidence presented here suggests that the results of bioassays with and without PBO can be used to predict the additional public health benefit of PBO LLINs. If there is a greater mortality in the PBO bioassay and the relative mortalities broadly agree with the red curve in *Figure 2B*, then *Figure 5B* can be used to predict the approximate number of cases that will be saved by switching from standard to PBO LLINs (for a given level of endemicity and LLIN coverage). Areas with 40–90% survival (10–60% mortality) in a non-PBO standard bioassay (of any type) should consider conducting PBO synergism bioassays to determine the suitability of PBO LLINs. We would suggest that either the WHO cone, WHO tube or CDC bottle assay (conducted in triplicate and averaged to improve precision) should be sufficient evidence to justify the need to switch to PBO LLINs.

The decision to recommend PBO nets over standard LLINs requires information on the relative cost effectiveness and affordability of PBO nets. If both net types cost the same and resistance has been detected then this work indicates that PBO LLINs should always be deployed as evidence suggests that they are always more effective. However, if PBO nets are more expensive, then cost effectiveness analysis will be required. The results of such analysis are likely to be context specific (depending on price, resistance level, endemicity and coverage) and interpreting them will require information on decision makers' willingness and ability to pay for additional effectiveness. In many situations, malaria control budgets are likely to be fixed and therefore switching to more expensive PBO LLINs may cause a reduction in overall bednet coverage. The impact of reduced coverage must therefore be off set against the benefits of introducing PBO nets, taking into consideration any additional factors such as changed programmatic costs, and equity issues.

Rapid deployment of new vector control products saves lives and gives incentives for industry to invest in new methods of vector control. New methods are likely to have a higher unit price than existing tools so it is important to be able to determine where and when they should be deployed in an efficient and transparent manner. Frameworks such as those presented here offer a relatively straightforward method of comparing two different products to determine whether the increased effectiveness justifies the higher unit price.

Much of the debate over the impact of pyrethroid resistance on LLIN effectiveness has focused on the loss of personal protection provided by new nets and does not fully take into account their community impact. A large body of evidence has shown how widespread use of LLINs can cause considerable community protection, both to those who use bednets and non-users (Killeen, 2007 and references therein). Therefore the community impact should be considered in any study investigating the consequences of pyrethroid resistance (*Briët et al., 2013*; *Killeen, 2014*), as any

**Table 2.** Inclusion and exclusion criteria used when conducting literature searches of published and grey literature. Pre-defined search string used are listed in *Figure 2—source data 1*.

| Inclusion criteria | Exclusion criteria |
|---|---|
| *General criteria across all meta-analyses* | |
| – Mosquito belong to the *A. gambiae* complex or *A. funestus* group<br>– Study conducted in Africa<br>– Bioassay must be of the standard dose for the particular pyrethroid (*WHO, 2013a*, *2013b*; *Brogdon, 2010*)<br>– Net must be a pyrethroid LLIN | – Studies which report percentage mortality but not the numbers tested / caught[†]<br>– Experimental hut trials which do not have adequate design to reduce bias (i.e. treatments arms were not rotated between huts; sleeper bias unaccounted for by preliminary testing; randomisation or rotation; huts were not cleaned between treatments)<br>– Experimental huts of the Ifakara design[‡] |
| *M1 –  Bioassay and experimental hut trial mortality* | |
| – Mosquito mortality measured in both an experimental hut study and separate bioassay (e.g. WHO tube assay, WHO cone assay, CDC bottle assay) | – Cone assays where the net had been washed |
| *M2 –  Impact of PBO in pyrethroid bioassays* | |
| – adult mosquito stage exposure to PBO | |
| *M3 –  Experimental hut trials of standard and PBO LLINS* | |
| – Study compares a combination LLIN (PermaNet 3.0 or Olyset Plus) with a conventional LLIN (PermaNet 2.0 or Olyset Net)[*]<br>– LLINs should be holed (Six 4 × 4 cm holes) | – Studies without both standard and PBO LLINs as non-parallel studies as studies from different sites may bias the difference between LLINs<br>– Trials without untreated control nets<br>– Studies which did not include feeding success |

[*] currently there are only two commercially available LLINs with PBO, PermaNet 3.0 (Vestergaard-Frandsen) and Olyset Plus (Sumitomo Chemicals Ltd). To limit the difference between LLIN types only nets made by the same manufacturer are directly compared.

[†] to increase the size of the bioassay dataset the authors of papers which failed to give sample sizes were contacted directly.

[‡] The probability that a mosquito will die in an experimental hut will depend on the hut design. To minimise the difference between studies, the most common design of hut is used, excluding the small number of studies which use the new Ifakara design (eg. *Okumu et al., 2013*).

reduction in mosquito killing is likely to increase malaria cases even in areas with mildly resistant mosquito populations where LLINs are still providing good personal protection. The evidence presented here suggests that high levels of pyrethroid resistance are likely to have a bigger public health impact than previously thought and therefore could represent a major threat to malaria control in Africa.

# Materials and methods

## Description of data

To generate results which are broadly applicable to all mathematical models were fit to data compiled by systematic meta-analyses of the published literature. Where possible meta-analyses were extended to the grey literature by including unpublished information. These include unpublished bioassay data from Liverpool School of Tropical Medicine, submissions to the World Health Organisation Pesticide Evaluation Scheme (WHOPES) and results from unpublished experimental hut trials (collated by contacting LLIN manufacturers Vestergaard-Frandsen and Sumitomo Chemicals Ltd). The meta-analyses followed the Preferred Reporting Items for Systematic Reviews and Meta-Analyses guidelines (*Moher et al., 2009*) for study search, selection and inclusion criteria though the study was not registered. The predefined inclusion criteria of each of the meta-analyses are presented in *Table 2* whilst the pre-defined search strings and the databases searched are outlined in full in *Figure 2—source data 1*. Extraction was done by N.L. into piloted forms. Study corresponding authors were contacted for raw data when this information was unavailable (all contacted investigators responded with the requisite information).

## Impact of pyrethroid resistance on LLIN mortality

To determine whether simple pyrethroid bioassays can be used to infer the outcome of experimental LLIN hut trials a meta-analysis (summarised as Meta-analysis 1, *M1*) was conducted to identify studies

**Table 3.** List of studies identified in meta-analysis *M1 - Predicting LLIN effectiveness from bioassay mortality*. Pre-defined search string used in the meta-analyses are listed in **Figure 2—source data 1** whilst raw data from are provided in **Figure 2—source data 2**.

| Study | Reference | Test | Country |
|---|---|---|---|
| 1 | *Ngufor et al. (2014a)* | WHO tube | Côte d'Ivoire |
| 2 | *Ngufor et al. (2014b)* | WHO tube | Benin |
| 3 | *Kitau et al. (2014)* | WHO tube | Tanzania |
| 4 | *Asale et al. (2014)* | WHO tube | Ethiopia |
| 5 | *Ngufor et al. (2014c)* | WHO tube | Burkina Faso |
| 6 | *Agossa et al. (2014)* | WHO tube | Benin |
| 7 | *Malima et al. (2013)* | WHO tube | Tanzania |
| 8 | *Adeogun et al. (2012b)* | WHO tube | Nigeria |
| 9 | *Koudou et al. (2011)* | WHO tube | Côte d'Ivoire |
| 10 | *Corbel et al. (2010)* | WHO tube | Benin, Burkina Faso, Cameroon |
| 11 | *Tungu et al. (2010)* | WHO tube | Tanzania |
| 12 | *Malima et al. (2008)* | WHO tube | Tanzania |
| 13 | *Kétoh (2016)* | WHO tube | Togo |
| 14 | *Toé (2015)* | WHO tube | Burkina Faso |

where both were carried out concurrently. To test whether this relationship changed with the population prevalence of insecticide resistance simple functional forms were fit to the raw data using a mixed-effect logistic regression (summarised as Relationship 1, *R1*). There has been an attempt to standardise bioassay and experimental hut trial procedures to enable data from different studies to be directly compared. These include using standard concentrations of insecticide, mosquito exposure time and mosquito husbandry in bioassays, hut design, trap type and the use of human baits in experimental hut trials. Nevertheless, some procedural discrepancies remain between studies, for example, in bioassays the age and sex of mosquitoes and how they were collected (e.g. F1 progeny of wild caught mosquitoes or wild caught larvae reared in insectary and tested as adults). These covariates and others (for example information on genetic markers associated with insecticide resistance), could be included within the analysis, though their addition would increase data needs of future studies and complicate the use of study results. Instead a mixed-effects binomial regression is adopted which allows mosquito mortality to vary at random between studies. This statistical method enables a wider selection of studies to be included within the analysis, produces more generalizable results and reduces problems caused by data autocorrelation. Mosquito mortality in an experimental hut trial is defined as the proportion of mosquitoes, which enter the hut which die, either within the hut or within the next 24 hr.

Meta-analysis 1 (*M1*) identified only 7 studies where concurrent bioassays and experimental hut trials were carried out (*Table 3*). Given the paucity of data results from all types of bioassay and mosquito species were combined and a simple, functional form was used to describe the relationship (the fixed-effect). Let $x$ denote the proportion of mosquitoes dying in a standard (non-PBO) pyrethroid bioassay then the population prevalence of pyrethroid resistance (expressed as a percentage, denoted $I$) is described by the following equation,

$$I = 100\,(1 - x). \tag{1}$$

Extending the notation of *Griffin et al. (2010)* the proportion of mosquitoes, which died in a hut trial is denoted $l_p$, where subscript $p$ indicates the net type under investigation, be it a no-net control hut ($p = 0$), a standard non-PBO LLIN ($p = 1$), or a PBO LLIN ($p = 2$). For a standard LLIN it is assumed to be explained by the equation,

$$\text{logit}(l_1) = \alpha_1 + \alpha_2(x - \tau), \tag{2}$$

**Table 4.** List of studies identified in meta-analysis *M2 - Estimating the impact of PBO in pyrethroid bioassays*. Bioassays run using laboratory strains are denoted. * Pre-defined search string used in the meta-analyses are listed in ***Figure 2—source data 1*** whilst raw data from are provided in ***Figure 2—source data 3***.

| Study | Reference | Test | Country |
|---|---|---|---|
| 1 | *Matowo et al. (2015)* | CDC tube | Tanzania |
| 2 | *Farnahm (1998)* | WHO tube | Uganda & Kenya |
| 3 | *Choi et al. (2014)* | WHO tube | Zambia & Zimbabwe |
| 4 | *Edi et al. (2014)* | WHO tube | Côte d'Ivoire |
| 5 | *Jones et al. (2013)* | WHO tube | Zanzibar |
| 6 | *Chouaibou et al. (2014)* | WHO tube | Côte d'Ivoire |
| 7 | *Koffi et al. (2013)* | WHO tube | Côte d'Ivoire |
| 8 | *Witzig et al. (2013)* | WHO tube | Chad |
| 9 | *Darriet and Chandre (2013)* | WHO tube | * |
| 10 | *Mawejje et al. (2013)* | WHO tube | Uganda |
| 11 | *Adeogun et al. (2012b)* | WHO tube | Nigeria |
| 12 | *Nardini et al. (2012)* | WHO tube | Nigeria |
| 13 | *Darriet and Chandre (2011)* | WHO tube | South Africa & Sudan |
| 14 | *Kloke et al. (2011)* | WHO cone | * |
| 15 | *Awolola et al. (2009)* | WHO tube | Mozambique |
| 16 | *Brooke et al. (2001)* | WHO tube | Nigeria |
| 17 | *N'Guessan et al. (2010)* | WHO tube | Mozambique |
| 18 | Ranson (2015) Personal Communication | WHO tube | Burkina Faso/Benin |
| 19 | Ranson (2015) Personal Communication | WHO tube | Chad colony |
| 20 | Morgan (2015) Personal Communication | WHO tube | Côte d'Ivoire |
| 21 | Ranson (2015) Personal Communication | WHO tube | Benin |
| 22 | Koudou & Malone (2015) Personal Communication | WHO cone | Côte d'Ivoire |
| 23 | PMI (2014). Personal Communication | CDC tube | Mali |
| 24 | *Toé (2015)* | WHO tube | Burkina Faso |
| 25 | *Abílio et al. (2015)* | WHO cone | Mozambique |
| 26 | *Riveron et al. (2015)* | WHO cone | Malawi |
| 27 | *Awolola et al. (2014)* | WHO cone | Nigeria |
| 28 | *Yewhalaw et al. (2012)* | WHO cone | Ethiopia |

where parameters $\alpha_1$ and $\alpha_2$ define the shape of the relationship and $\tau$ is a constant used to centre data to aid the fitting process. More sophisticated functional forms could be used for *R1* (***Equation [2]***) though they were not currently warranted given the limited dataset. Let $N_p$ indicate the number of mosquitoes entering a hut in an experimental hut trial. If the number of these mosquitoes which enter the hut and subsequently die ($L_1$) follows a binomial distribution then parameters $\alpha_1$ and $\alpha_2$ can be estimated for a non-PBO net by fitting the following equation to *M1*,

$$L_1 \sim \mathrm{B}(l_1, N_1) + \epsilon_\alpha. \tag{3}$$

The random-effects component is included by allowing mortality to vary at random between sites by adding the error term $\epsilon_\alpha$ which has a mean of zero and a constant variance.

## Estimating the impact of PBO on pyrethroid induced mortality

The number of experimental hut trials investigating the difference between standard and PBO nets is limited. Instead a meta-analysis of all bioassay data investigating the impact of PBO on pyrethroid induced mosquito mortality is undertaken incorporating all published and unpublished literature (*M2, Table 4*). Bioassay mortality can be influenced by a multitude of factors including assay type, temperature and relative humidity (*Kleinschmidt et al., 2015*). To account for this difference between studies, the relationship between the benefit of adding PBO and the population prevalence of pyrethroid resistance was estimated using a mixed-effect logistic regression (*R2*). Preliminary analysis suggests that the shape of the relationship is relatively complex and cannot simply be described by the use of a standard linear function typically used in regression. Since the added benefit of PBO in a given population will ultimately be determined by the shape of this relationship a variety of different functional forms are tested statistically. It was initially intended to include the type of assay used (e.g. WHO tube assay, WHO cone assay or CDC bottle assay) as an additional fixed effect, though the paucity of data (especially comparing bioassay mortality to experimental hut trial mortality) meant that data from all assays were combined and this covariate was excluded. As the same type of assay are used for both non-PBO and PBO tests this should not bias the results and will generate recommendations that are generalizable across all three assay types. The proportion of mosquitoes killed by pyrethroid insecticide in a bioassay with the addition of PBO is denoted $f$ and is given by the equation:

$$\text{logit}(f) = \beta_1 + \frac{\beta_2(x-\tau)}{1+\beta_3(x-\tau)} \qquad [4]$$

where $x$ is the proportion of mosquitoes dying in a non-PBO bioassay, parameters, $\beta_1, \beta_2$ and $\beta_3$ define the shape of the relationship and $\tau$ is a constant supporting the fitting process (this relationship is referred to as *R2*). Let $A_i$ be the number of mosquitoes used in a bioassay and $D_i$ the number which died, with subscript $i$ denotes whether or not PBO was added to the bioassay ($i$ = 1 pyrethroid alone, $i$ = 2 pyrethroid plus PBO). If it is assumed that the number of mosquitoes that die in the bioassay follows a binomial distribution then parameters, $\beta_1, \beta_2$ and $\beta_3$ can be estimated by fitting the following equations to the dataset from (*M1*),

$$D_1 \sim \text{B}(x, A_1) + \epsilon_\beta, \qquad [5]$$

$$D_2 \sim \text{B}(f, A_2) + \epsilon_\beta. \qquad [6]$$

Parameter $\epsilon_\beta$ represents a normally distributed random error with a mean of zero and a constant variance and is estimated from the fitting procedure.

**Table 5.** List of studies identified in meta-analysis *M3 - Estimating the impact of PBO in experimental hut trials*. Pre-defined search string used in the meta-analyses are listed in *Figure 2—source data 1* whilst raw data from published studies are provided at doi:10.5061/dryad.13qj2.

| Study | Reference | Country |
|---|---|---|
| 1 | *Pennetier et al. (2013)* | Benin, Cameroon |
| 2 | *Adeogun et al. (2012a)*) | Nigeria |
| 3 | *Corbel et al. (2010)* | Benin, Burkina Faso, Cameroon |
| 4 | *Tungu et al. (2010)* | Tanzania |
| 5 | *N'Guessan et al. (2010)* | Benin |
| 6 | Kétoh *et al.*, Unpublished | Togo |
| 7 | Tungu *et al.*, Personal Communication | Tanzania |
| 8 | Toé *et al.*, Personal Communication | Burkina Faso |

## Predicting the added benefit of PBO LLINs in experimental hut trials

Relationships *R1* and *R2* can be used to predict the effectiveness of PBO LLINs in experimental hut trials. When bioassay data are unavailable the current population prevalence of insecticide resistance can be predicted from mosquito mortality measured in a standard LLIN experimental hut trial by rearranging *Equation [2]*,

$$\hat{x} = \left[ \left( \frac{\exp(l_1)}{1 - \exp(l_1)} \right) - \alpha_1 \right] / \alpha_2 + \tau, \qquad [7]$$

where the section in round brackets is the inverse logit function. This equation together with *Equations [2] and [4]* can be then used to predict the relationship between hut trial mortality in standard and PBO LLINs for a range of areas with different levels of pyrethroid resistance using the following steps (a) to (c) below.

 a. For a range of values of $l_1$ (proportion of mosquitoes which died in a standard LLIN hut trial) generate the predicted population prevalence of mosquito mortality in a bioassay expected in the population $\hat{x}$ using *Equation [7]*.

 b. Use $\hat{x}$ to predict pyrethroid induced mortality in a bioassay with PBO $\hat{f}$ given the current population prevalence of pyrethroid resistance (i.e. substitute $\hat{x}$ for $x$ in *Equation [4]*).

 c. Convert the expected mortality in a bioassay $\hat{f}$ into the expected mortality in a PBO LLIN hut trial (i.e. substitute $\hat{f}$ for $\hat{x}$ in *Equation [2]*).

To test the predictive ability of *R1* and *R2* a third meta-analysis was carried out for all experimental hut trials which directly compare standard and PBO pyrethroid LLINs (*M3*, *Table 5*). The accuracy of these predictions can then be examined by comparing them visually (*Figure 2C*) or statistically using an Anaylsis of Variance.

## Quantifying the impact of standard and PBO LLINs in the presence of insecticide resistance

The impact of insecticide resistance on mosquito interactions with LLINs is systematically investigated by analysing the experimental hut trials identified in *M3*. Restricting the analysis to the two most commonly used standard LLINs minimises the inter-study variability and allows the different behaviours of mosquitoes exposed to standard and PBO LLINs to be directly assessed. Following a widely used transmission dynamics model of malaria (*Griffin et al., 2010*; *Walker et al., 2015*) it is assumed that an LLIN can alter a host-seeking mosquito behaviour in one of three ways: firstly it can deter a mosquito from entering a hut (an exito-repellency effect); secondly the mosquito can exit the hut without taking a bloodmeal; and thirdly it could kill a mosquito (with the mosquito either being fed or unfed). A mosquito that isn't deterred, exited or killed will successfully blood-feed and survive. The public health benefit of LLINs depends not only on their initial effectiveness but also on how the properties of the net changes over its life-time. The ability of a net to kill a mosquito will decrease over time as the quantity of insecticide active ingredient declines. The non-lethal protection provided by the LLIN may also decrease with the decay of the active ingredient and the physical degradation of the net (i.e. the acquisition of holes). It is assumed that the underlying difference in hut trial mortality between sites for standard LLINs is caused by the mosquito population having a different population prevalence of pyrethroid resistance. Pyrethroid resistance may also influence the relative strength of LLIN deterrence and exiting and it is important to characterise these modifications of behaviour as they contribute substantially to the population level impact of mass LLIN distribution. Visual inspection of these data indicates that mosquito deterrence and exiting can be described by the degree of mosquito mortality seen in the same hut trial.

The proportion of mosquitoes not deterred from entering a hut by the LLIN is estimated using $m_p$, the ratio of the number of mosquitoes entering a hut with an LLIN ($N_1$ or $N_2$) to the number entering a hut without a bednet ($N_0$, here assumed to be the same as a hut with an untreated bed net). A statistical model is used to determine whether there is an association between the number of mosquitoes entering a hut with a standard LLIN and the proportion of mosquitoes which die when they do (which is assumed to be a proxy for mosquito susceptibility, i.e. $m_1$ is described by $l_1$ and $m_2$ is described by $l_2$). It is assumed that the shape of the relationship between the proportion of

mosquitoes entering a hut with an LLIN relative to a hut with an untreated net (1-deterrence) and mortality is described by the flexible third order polynomial,

$$m_p = 1 - \left[ \delta_1 + \delta_2 (l_p - \tau) + \delta_3 (l_p - \tau)^2 \right] \quad [8]$$

$$N_p \sim \mathrm{N}(m_p N_0, \delta_4) \quad [9]$$

Though there is no *a priori* reason to assume an inflection point in the relationship between $m_p$ and $l_p$ the polynomial function is chosen as it is highly flexible and would allow such a curve should it exist (which is necessary given the variability in the raw data). The shape parameters $\delta_1$, $\delta_2$ and $\delta_3$ are estimated assuming that the number of mosquitoes caught has a normal distribution (verified using a and deterrence is allowed to vary at random between sites (with variance $\delta_4$).

The proportion of mosquitoes entering the hut which exit without feeding is denoted $j_p$ whilst the proportion which successfully feed upon entering is $k_p$. Once entered the hut mosquitoes have to either exit, die or successfully feed (i.e. $1 = j_p + l_p + k_p$). Visual inspection of these data indicates that $k_p$ increases with decreasing mortality at an exponential rate (*Figure 3C*). Therefore, if the number of mosquitoes which feed and survive ($S_p$) follows a binomial distribution then,

$$S_p \sim \mathrm{B}(k_p, N_p) + \epsilon_\theta \quad [10]$$

$$k_p = \theta_1 exp\left[\theta_2 (1 - l_p - \tau)\right] \quad [11]$$

where $\theta_1$ and $\theta_2$ determine the shape of the relationship and $\epsilon_\theta$ is a normally distributed random error which varies between sites.

## Parameterising transmission dynamics model

Estimates of $j_p$, $l_p$ and $m_p$ can be used to determine the proportion of mosquitoes repeating (a combination of deterrence and exiting, $r_{p0}$), dying ($d_{p0}$) and feeding successfully ($s_{p0}$) during a single feeding attempt in a hut with a new LLIN relative to those successfully feeding in a hut without an LLIN (i.e. $p$=1 or 2),

$$r_{p0} = \left(1 - \frac{k_p'}{k_0}\right)\left(\frac{j_p'}{j_p' + l_p'}\right) \quad [12]$$

$$d_{p0} = \left(1 - \frac{k_p'}{k_0}\right)\left(\frac{l_p'}{j_p' + l_p'}\right) \quad [13]$$

$$s_{p0} = \frac{k_p'}{k_0} \quad [14]$$

where $j_p = 1 - l_p - k_p$, $j_p' = m_p j_p + (1 - m_p)$, $k_p' = m_p k_p$ and $l_p' = m_p l_p$ (*Griffin et al., 2010*). Not all mosquitoes which enter a house will successfully feed even if there are no vector control interventions inside. The experimental hut trials used in this analysis did not include a no-net control ($k_0$) so historical studies are used for this parameter (*Curtis et al., 1996*; *Lines et al., 1987*). Though theoretically $s_{p0}$ could have values > 1 for practical purposes, it is constrained between zero and one as on average mosquitoes entering a hut with an LLIN are less likely to feed than a mosquito entering a hut without a bednet (as shown by all estimates of $k_p$ being substantially lower than $k_0$, see *Figure 3C* and *Table 6*). The majority of experimental hut trials in *M3* are in areas where the dominant vector is *A. gambiae s.s.* and no studies were conducted in areas with *A. funestus*. As there is insufficient information to generate these functions for each species separately it is assumed that the relationship between $r_{p0}$, $s_{p0}$ and $d_{p0}$ are consistent across all species. The average effectiveness of LLINs in an entirely susceptible mosquito population identified in *M3* is slightly higher than those analysed by *Griffin et al. (2010)* which included a wider range of LLIN types. Values of $m_p$ (the propensity of mosquitoes to enter a hut with an LLIN relative to one without) greater than one are truncated at

**Table 6.** Parameter definitions and fitted values. Unless otherwise stated, all other parameters used were taken from *Griffin et al. (2010)*. Some parameters are mosquito species-specific whilst others are constant within a species complex (denoted *) or universal (species independent $).

| Parameter definitions | | *Anopheles gambiae* s.s. | *Anopheles arabiensis* | *Anopheles funestus* |
|---|---|---|---|---|
| $x$ | proportion mosquitoes dying in a discriminating dose pyrethroid bioassay | - | | |
| $I$ | population prevalence of pyrethroid resistance (percentage survival) estimated using $x$ (*Equation [1]*) | - | | |
| $p$ | net type under investigation in experimental hut trials: untreated ($p = 0$); standard LLIN ($p = 1$); PBO LLIN ($p = 2$). | - | | |
| $d_p$ | probability a mosquito dies during single feeding attempt (*Equation [18]*) | Estimated from parameters below | | |
| $r_p$ | probability a mosquito exits the hut during single feeding attempt (*Equation [17]*) | Estimated from parameters below | | |
| $s_p$ | probability a mosquito feeds during single feeding attempt (*Equation [19]*) | Estimated from parameters below | | |
| $N_p$ | the number of mosquitoes entering a hut with net type $p$ (*Equation [3]*) | - | | |
| $m_p$ | proportion of mosquitoes entering a hut with an LLIN to relative to a hut with an untreated bed net ($N_p/N_0$, *Equation [8]*)$^\$$. | $\delta_1 = 0.071$ $\delta_2 = 1.26$ $\delta_3 = 1.52$ | | |
| $l_p$ | proportion of mosquitoes that enter a hut with net type $p$ that die (*Equation [2]*)$^\$$ | $\alpha_1 = 0.63$ $\alpha_2 = 4.00$ | | |
| $k_p$ | proportion of mosquitoes that enter a hut with net type $p$ that successfully feed and survive (*Equation [11]*)$^\$$ | $\theta_1 = 0.02$ $\theta_2 = 3.32$ | | |
| $j_p$ | proportion of mosquitoes that enter a hut with net type $p$ that exit without feeding | $1 - l_p - k_p$ | | |
| $\gamma_p$ | rate of decay in insecticide activity (in washes) for net type $p$ (*Equation [16]*)$^\$$ | $\mu_p = -2.36$ $\rho_p = -3.05$ | | |
| $f$ | proportion of mosquitoes killed in pyrethroid + PBO bioassay (*Equation [4]*)* | $\beta_1 = 3.41$, $\beta_2 = 5.88$, $\beta_3 = 0.78$ | | $\beta_1 = 2.53$ $\beta_2 = 0.89$ |
| $\tau$ | constant used to centre the data to aid the fitting process | 0.5 | | |

*Relevant parameters previously estimated by* **Griffin et al. (2010)**[†] *and* **Walker et al. (2015)**[‡]

| | | | | |
|---|---|---|---|---|
| $k_0$ | proportion of mosquitoes that enter a hut with no bednet that successfully feed$^\$$ | 0.70[†] | | |
| $H_y^s$ | insecticide activity half-life in years for a susceptible mosquito population $^\$$ | 2.64[†] | | |
| $r_M$ | proportion of mosquitoes which exit the hut when LLIN has no insecticidal activity | 0.24[†] | 0.24[‡] | 0.24[†] |
| - | mean life expectancy (days) | 7.6[†] | 7.6[‡] | 8.9[†] |
| - | proportion blood meals taken on humans without LLINs (human blood index) | 0.92[†] | 0.71[‡] | 0.94[†] |
| - | proportion of bites taken on humans whilst they are in bed | 0.89[†] | 0.83[‡] | 0.90[†] |

one as there is insufficient evidence to justify that mosquitoes preferentially enter huts with LLINs (in part because the number of studies with very low mortality are low and the metric has high measurement error).

## Decay in LLIN efficacy over time

The ability of a net to kill a mosquito will decrease over time as the quantity of insecticide active ingredient declines. The non-lethal protection provided by the LLIN may also decrease with the decay of the active ingredient and the physical degradation of the net (i.e. the acquisition of holes). To fully capture the loss of efficacy of an LLIN requires a net durability survey to be carried out over multiple years. To our knowledge, no durability studies have been published in areas of high pyrethroid resistance nor using the new generation of LLINs with the addition of PBO. In the absence of these data, we use the results from experimental hut trials that washed the net prior to its use. These experimental huts give some indication of how mosquitoes react to the change in insecticide concentration, though they do not provide information on the physical durability of the net (as holes in the net are artificially generated). For simplicity and following (*Griffin et al., 2010*) it is assumed that

the killing activity of pyrethroid over time (the half-life in years, denoted $H_y$) is proportional to the loss of morbidity caused by washing (the half-life in washes, $H_w$). A prior estimate of the half-life in years (*Mahama et al., 2007*) from a durability study of a non-PBO LLIN with susceptible mosquitoes ($H_y^s$) is then used to reflect changes caused by pyrethroid resistance by,

$$H_y = H_w / H_w^s H_y^s \qquad [15]$$

where superscript $s$ indicates the half-life in a fully susceptible mosquito population (i.e. $l_1 = 1$). Note that if the newer PBO nets have better durability than standard LLINs then this will under estimate their additional benefit. Following *Griffin et al., 2010* it is assumed that the activity of the insecticide decays at a constant rate according to a decay parameter $\gamma_p$, which is related to the half-life by $H_w = \ln(2)/\gamma_p$. To test whether the rate of decay changes with $l_p$ (i.e. mosquito mortality caused by new standard and PBO LLINs) the following equation was fit to *M3*,

$$\mathrm{logit}(\gamma_p) = \mu_p + \rho_p(l_p - \tau). \qquad [16]$$

Shape parameters $\mu_p$ and $\rho_p$ are allowed to vary between net types. The proportion of mosquitoes repeating due to the LLIN decreases from a maximum, $r_{p0}$, to a non-zero level $r_M$, reflecting the protection still provided by an LLIN that no longer has any insecticidal activity. For simplicity, it is assumed that the rate of decay from $r_{p0}$ to $r_M$ is given by $\gamma_p$ (as the degradation of the net over time is unlikely to be recreated by washing). The full equations for the proportion of mosquitoes repeating, dying and successfully feeding at time $t$ following LLIN distribution ($r_p$, $d_p$ and $s_p$, respectively) is given by,

$$r_p = (r_{p0} - r_M)\exp(-\gamma_p t) + r_M \qquad [17]$$

$$d_p = d_{p0}\exp(-\gamma_p t) \qquad [18]$$

$$s_p = 1 - r_p - d_p. \qquad [19]$$

## Fitting procedure

All models were fit using a Markov chain Monte Carlo sampling algorithm implemented in the programme OPENBUGS (*Lunn et al., 2009*). This Bayesian method enabled measurement error to be incorporated in both the dependent and independent variables according to the number of mosquitoes sampled (both in bioassays and hut trials). Uninformative priors were used for all parameters with the exception of the random effects variance parameters which were constrained to be positive (though were still uninformative, see *Source code 1* in the Supplementary Information for a full list of priors). Three Markov chains were initialized to assess convergence and the first 5000 Markov chain Monte Carlo iterations were discarded as burn in. Convergence was assessed visually and a total of 10,000 iterations were used to derive the posterior distribution for all parameters and to generate 95% Bayesian credible interval estimates for model fits. The models were compared using the deviance information criterion (DIC) where the smaller value indicates a better fit, and a difference of five deviance information criterion units is considered to be substantial (*Spiegelhalter et al., 2002*). *Equations [8] to [19]* were fit simultaneously to *M3* enable the impact of washed nets to contribute to the relationship between $r_p$, $d_p$ and $s_p$, through the decay function, $\gamma_p$, doubling the number of datapoints in the analysis. A direct comparison between net types is beyond the scope of this study. Only one study compared PermaNet 2.0 and PermaNet 3.0 at the same time and place as Olyset and Olyset Plus and this study did not conduct hut trials with washed LLINs. As the different nets were tested in areas with different levels of pyrethroid resistance (in part because the low overall number of studies) then the impact of resistance and net type cannot currently be disentangled.

## Predicting the public health impact of insecticide resistance

The public health benefit of PBO-LLINs will depend on the epidemiological setting in which they are deployed. This includes the baseline characteristics of the setting (e.g. mosquito species, abundance and seasonality), history of malaria control interventions (e.g. prior use of bednets, management of clinical cases) and prevalence of insecticide resistance. The rate at which pyrethroid resistance has

evolved is highly uncertain. It is likely that it first became evident through its use in agriculture and the relative contribution of vector control to the selection of resistance is unknown and will vary between sites. This makes it impossible to recreate the spread of resistant phenotypes in a particular setting and predict its cumulative public health impact without detailed longitudinal studies spanning decades (which do not exist for malaria endemic regions). Instead the impact of pyrethroid resistance is estimated by assuming it arrives instantaneously at a given level. To generate a broadly realistic history of LLIN usage it is assumed that LLINs were introduced at a defined coverage at year zero and redistributed every three years to the same percentage of the human population (*Figure 1*). The mosquito population is assumed to be either *A. gambiae* s.s., *A. arabiensis* or *Anopheles funestus* (the three major vectors in Africa) which are entirely susceptible to pyrethroids up until year 6 when pyrethroid resistance arrives instantaneously. The public health impact of resistance is then measured over the subsequent three years (the average clinical incidence or entomological inoculation rate (EIR) between the years 6 and 9) and compared to a population where resistance did not arise. The impact of PBO LLINs is predicted by introducing them into the resistant population at the year 9 and then measuring over the subsequent 3 years. For simplicity, it is assumed that there is perennial transmission, no other type of vector control and that once introduced pyrethroid resistance remains constant. Though perennial transmission is unrealistic it is necessary in order to produce simple guidelines (as there is a very high number of combinations of seasonal patterns, relative mosquito species abundance and timings of LLIN distribution campaigns). A sensitivity analysis with more realistic seasonal patterns shows the change in clinical incidence compared to the perennial transmission is relatively minor, in part because the LLINs are used over 3 yearly cycles and their decay in effectiveness is relatively slow. LLINs are initially distributed at time zero at random (i.e. there was no targeting to those with the highest infection) and from then on the same people receive them every campaign to ensure that coverage remains at the defined level (i.e. the number of people with an LLIN would go up if the distribution was random each round). Realistic usage patterns are adopted to reflect higher coverage immediately after LLIN distribution. No other vector control is incorporated whilst 35% of clinical cases are assumed to receive treatment, 36% which receive an ACT (estimated by averaging across Africa using data collated by *Cohen et al., [2012]*). A full list of the parameters, their definitions and estimated values are given in *Table 6* whilst all other parameters are taken from *Griffin et al. (2010)* and *White et al. (2011)*.

To investigate how the uncertainty in mosquito behaviour and the impact of PBO influence model predictions, a full sensitivity analysis is carried out for the parameters determining LLIN efficacy. A thousand parameter sets for $\alpha_1$, $\alpha_2$, $\beta_1, \beta_2$, $\delta_1$, $\delta_2$, $\theta_1$, $\theta_2$, $\mu_p$ and $\rho_p$ are sampled from the posterior distribution and are used to generate a range of possible values for $r_{p0}$, $s_{p0}$, $d_{p0}$ and $\gamma_p$ (*Figure 4—figure supplement 5*). This allows uncertainty in all measurements (such as the relationship between resistance and hut trial mortality) to be propagated throughout the equations. These parameter sets are then included as runs within the full transmission dynamics model to unsure the full uncertainty in these data is represented and the 95% credible intervals for model outputs are then shown.

### Source data

*Figure 2—source data 1–3*. Figure 2—source data 4 is hosted on Dryad (doi: 10.5061/dryad.13qj2)

## Acknowledgements

TSC would like to thank the IVCC (Innovative Vector Control Consortium) and the UK Medical Research Council (MRC) / UK Department for International Development (DFID) under the MRC/DFID Concordat agreement. The financial support of the European Union Seventh Framework Programme FP7 (2007–2013) under grant agreement no 265660 *AvecNet* is gratefully acknowledged. NL was supported by an ISSF Grant from the Wellcome Trust.

# Additional information

## Funding

| Funder | Grant reference number | Author |
|---|---|---|
| Medical Research Council | | Thomas S Churcher |
| Department for International Development | | Thomas S Churcher |
| Innovative Vector Control Consortium | | Thomas S Churcher |
| Wellcome Trust | ISSF Grant | Natalie Lissenden |
| European Research Council | 265660 | Hilary Ranson |

The funders had no role in study design, data collection and interpretation, or the decision to submit the work for publication.

## Author contributions

TSC, Conception and design, Acquisition of data, Analysis and interpretation of data, Drafting or revising the article; NL, Acquisition of data, Drafting or revising the article; JTG, Analysis and interpretation of data, Drafting or revising the article; EW, HR, Conception and design, Drafting or revising the article

## Author ORCIDs

Thomas S Churcher, http://orcid.org/0000-0002-8442-0525

# Additional files

## Supplementary files

• Source code 1. All OPENBUGS code used to fit the functional relationships between variables are included below.

## Major datasets

The following dataset was generated:

| Author(s) | Year | Dataset title | Dataset URL | Database, license, and accessibility information |
|---|---|---|---|---|
| Churcher TS, Lissenden N, Griffin JT, Worrall E, Ranson H | 2016 | Data from: The impact of pyrethroid resistance on the efficacy and effectiveness of bednets for malaria control in Africa | http://dx.doi.org/10.5061/dryad.13qj2 | Available at Dryad Digital Repository under a CC0 Public Domain Dedication |

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
