## [Decision Letter]

Thank you for resubmitting your work entitled "The impact of pyrethroid resistance on the efficacy and effectiveness of bednets for malaria control" for further consideration at *eLife*. Your revised article has been favorably evaluated by Prabhat Jha (Senior editor), a Reviewing editor, and three reviewers.

There are some issues that need to be addressed before acceptance, as outlined below:

Reviewer 1 raises many important points for further clarification of the model and reviewer 2 some clear guides for improvement in terms of the justification/motivation of the models, its extrapolation to Africa and the overall messaging of the paper. The specific concerns are as follows:

*Reviewer #1:*

This paper combines three meta-analyses of bioassays and hut trials with analysis based on a transmission dynamic model to explore how different levels of pyrethroid resistance might be expected to affect malaria incidence and the potential benefits of switching to nets containing PBO.

Clearly this work addresses a very important public health problem. I believe the broad class of methods used are appropriate, though further justification for some specific assumptions is needed (see below). The work claims to offer substantial new insights, in particular highlighting the importance of even low levels of resistance which has the potential to change the way pyrethroid resistance is thought about. I don't know the LLIN literature well enough to assess the accuracy of this claim, but it sounds plausible.

The paper is clearly written, the results are explained very well, and the figures clearly convey the key findings. The methods are also well-described (though will be even clearer once code has been made available as is planned). There is one area that I think needs to be improved: while the methods describe clearly what was done, it is not always clear why. In particular, I found the reasons behind a number of modelling choices described in the subsections “Quantifying the impact of standard and PBO LLINs in the presence of insecticide resistance” and “Parameterising transmission dynamics model” opaque. For example, why use a 3rd order polynomial when looking at LLIN deterrence? Are there reasons for thinking there should be two change-points? To what extent are the assumptions of normally distributed errors ([Disp-formula equ9]) justified? Can the data motivating the choice of [Disp-formula equ10 equ11] be shown in a technical appendix? Motivation of [Disp-formula equ12 equ13 equ14] is also lacking: taking the simplest ([Disp-formula equ13]) it is unclear why s_p0 (the proportion of mosquitoes feeding successfully) should depend on both m_p (the ratio of the number of mosquitoes entering a hut with a LLIN to the number entering a hut without a bednet), and on k_p (the proportion which enter and successfully feed a p-treated hut), given that the latter already includes entering. Moreover, it is claimed that s_p0 is a proportion, so should be constrained between 0 and 1, but if m_p = 1 (so bednets have no effect on entering) then s_p0=k_p/k_0, which is a ratio of two proportions rather than a proportion itself (for plausible parameters values it presumably will be below 1, but it is not necessarily so). I therefore think further motivation is required, which should ideally include some graphical assessments of model assumptions (3rd order polynomial, normally distributed errors, relationship between k_p and mortality) which could go in the appendix.

It is also good practice in any Bayesian analysis to explicitly show posterior distributions of model parameters or at the very least to summarise CrIs for these.

*Reviewer #1 (Additional data files and statistical comments):*

It would be useful to provide code and data used in meta analyses – and the statistical submission form states that these will be made available.

*Reviewer #2:*

The authors have used three small datasets (all that is available) and many assumptions to model the impact of insecticide resistance, defined using insecticide bioassay results, on malaria incidence in Africa in a defined range of scenarios. Their scope is ambitious and this work brings together several separate studies to incorporate resistance into an existing malaria model. The work builds on earlier studies and is novel and interesting, however, the results need to be caveated carefully to recognise the limitations of the datasets used and the assumptions made. It also needs to be clear that this work makes predictions for a limited range of scenarios.

The work uses a series of meta-analyses to show that insecticide bioassay results can be used to predict the impact of resistance on malaria incidence, however, the final model has not been validated using African locations that were not included in the meta-analyses datasets where values are known for both malaria incidence and resistance as measured by an insecticide bioassay.

The data used, the mosquito species included and the range of slide prevalence values considered are all specific to scenarios found in Africa but nowhere in the paper is this limitation mentioned. The implication from the title and throughout the manuscript is that this is a generalizable analysis of pyrethroid resistance but it is in fact only applicable in Africa, and only in certain scenarios within subsaharan Africa.

Throughout the Results and Discussion there are sentences that look like statements of fact but they are in fact predictions within the limits of certain scenarios defined by the authors and predicated on analysis of a limited dataset and many assumptions, for example, "For the *An. gambiae* complex PBO had the greatest benefit in mosquito populations with intermediate levels of pyrethroid resistance", "The probability that a mosquito will feed on someone beneath a LLIN only increases at high levels of pyrethroid resistance", and so on. These statements need to be presented as predictions, and it would also be useful if the Results could start with a summary of the scenarios covered and assumptions made (see below). In Results, the authors state "The numbers of mosquitoes deterred from entering the experimental hut substantially decreases in areas of higher pyrethroid resistance" – this is a strong statement but when you look at Figure 3 you can see that the credible intervals are large ("substantial" even) and this statement needs to caveated appropriately.

Where values are given, these are not bounded by any intervals, for example in Results "causing up to 200 additional cases per 1000" or "where over 500 cases per 100 people can be prevented each year". A range or intervals are needed to give an idea of uncertainty.

Meta-analyses 1 and 2 were compared to the observed results collected for the third meta-analysis but this comparison was visual only with no formal analysis or validation (end of subsections “Added benefit of PBO” and” Predicting the added benefit of PBO LLINs in experimental hut trials”).

The paper as a whole, and the Methods in particular, is a long and dense read and would benefit from summaries for biomedical/entomological readers who are not mathematical modellers, whilst still retaining important details about the scenarios modelled and the assumptions made. In particular, summaries aimed at these readers at the beginning of both the Methods and Results would be hugely helpful for *eLife*'s broader readership.

Figure 1 is cited in the Introduction and seems to be a key result but is not explained in the Results at all or discussed. No credible intervals are included. It shows model predictions for the scenario where bioassay results give 20% mortality but it would be interesting to see the results for other mortality/survival rates that are often found in the region. The green dashed line in 1A shows 10% parasite prevalence but it is unclear why. The starting prevalence is >20% and then it drops after control but come up to 10% every three years so presumably this is the setting the green line refers to? The legend says the black line shows the situation with no resistance and the red line shows the situation if resistance arrives at Y6, but the red lines starts before Y0 and the black line doesn't start until Y6.

There are a lot of assumptions made by this work but it is unclear how they have been justified and which ones have been subject to sensitivity analyses in the context of the results presented here, i.e. the impact of the prevalence of resistance on malaria incidence. The assumptions made include: Assumed mosquito deterrence and exiting can be described by the degree of mosquito mortality seen in the same hut trial; Assumed the relationship between deterrence/exiting, feeding successfully and dying is consistent across all species; Assumed washing nets gives the same results as a durability study; Assumed the activity of the insecticide decays according to a given formula that includes half-life in washes; Assumed resistance arises spontaneously, and after six years of LLINs use; Assumed LLINs are re-distributed every three years; Assumed transmission is perennial; Assumed there is no other vector control (and presumably no other non-vector related pyrethroid use); Assumed resistance remains constant after arising; Assumed 35% clinical cases are treated of which 36% receive ACT); Incorporated assumptions/estimates used previously by the same group that are not all given here; Assumed physiological resistance has no effect on the vectorial capacity of individual mosquitoes.

*Reviewer #2 (Additional data files and statistical comments):*

I am not a mathematical modeller and assume that this manuscript has also gone to reviewers who can comment on the modelling work in more depth.

The authors propose to make the data used in the meta-analysis available via Dryad if they have previously been published. This is reasonable but some of the unpublished datasets were provided by the authors of this paper and so should also be included in data deposit.

I am not qualified to comment on whether the source code provided would allow a reader to repeat this work but this is important.

*Reviewer #3:*

This is a well written, well-designed and important manuscript.

The observation that bioassay survival can be used as a quantitative test to assess the level of pyrethroid resistance is an important one. As the authors note bioassays are a crude tool but they can be potential important on a programmatic level.

The observation that LLINs provide protection until high levels of resistance is important from a policy perspective and contributes to understanding a poorly studied topic.

The model is statistically sound and the authors must be commended on sharing their code for full transparency.

The final recommendation of the benefits of PBO nets has important policy implications for areas high in resistance.

---

## [Author Response]

*There are some issues that need to be addressed before acceptance, as outlined below:*

*Reviewer 1 raises many important points for further clarification of the model and reviewer 2 some clear guides for improvement in terms of the justification/motivation of the models, its extrapolation to Africa and the overall messaging of the paper. The specific concerns are as follows:*

*Reviewer #1:*

*This paper combines three meta-analyses of bioassays and hut trials with analysis based on a transmission dynamic model to explore how different levels of pyrethroid resistance might be expected to affect malaria incidence and the potential benefits of switching to nets containing PBO.*

*Clearly this work addresses a very important public health problem. I believe the broad class of methods used are appropriate, though further justification for some specific assumptions is needed (see below). The work claims to offer substantial new insights, in particular highlighting the importance of even low levels of resistance which has the potential to change the way pyrethroid resistance is thought about. I don't know the LLIN literature well enough to assess the accuracy of this claim, but it sounds plausible.*

*The paper is clearly written, the results are explained very well, and the figures clearly convey the key findings. The methods are also well-described (though will be even clearer once code has been made available as is planned). There is one area that I think needs to be improved: while the methods describe clearly what was done, it is not always clear why. In particular, I found the reasons behind a number of modelling choices described in the subsections “Quantifying the impact of standard and PBO LLINs in the presence of insecticide resistance” and “Parameterising transmission dynamics model” opaque. For example, why use a 3rd order polynomial when looking at LLIN deterrence? Are there reasons for thinking there should be two change-points?*

We agree that the choice of curves could be more thoroughly explained. There is no a priori reason to assume that there is a change-point though the polynomial was chosen as it provides complete flexibility for the shape of the relationship given that there is no obvious pattern to these data. This flexibility indeed shows that some of the 95% credible interval curves do have a change point though more studies would be required to thoroughly evaluate this. We have changed the text accordingly to give further justification:

“Though there is no a priori reason to assume an inflection point in the relationship between m_p and l_p the polynomial function is chosen as it is highly flexible and would allow such a curve should it exist (which is necessary given the variability in the raw data).”

*To what extent are the assumptions of normally distributed errors ([Disp-formula equ9]) justified?*

The assumption of normally distributed errors was investigated using a quantile-quantile plot which indicates that these data is adequately described using a normal distribution. This figure has been included in Figure 3—figure supplement 1 whilst caption says:

“(A) shows a normal quantile-quantile plot for the residuals of the data for the relationship between deterrence and mosquito survival in experimental hut trials (Figure 3, equation). The linearity of the residuals (the proximity of the blue dots to the red dotted line) indicate that the error in these data are adequately described by the normal distribution (see [Disp-formula equ9])”.

Can the data motivating the choice of [Disp-formula equ10 equ11] be shown in a technical appendix?

The data motivating the choice of [Disp-formula equ11] is shown in Figure 3. We have made this clear in the manuscript by including the following text:

“Visual inspection of these data indicates that k_p increases with decreasing mortality at an exponential rate (Figure 3).”

*Motivation of [Disp-formula equ12 equ13 equ14] is also lacking: taking the simplest ([Disp-formula equ13]) it is unclear why s_p0 (the proportion of mosquitoes feeding successfully) should depend on both m_p (the ratio of the number of mosquitoes entering a hut with a LLIN to the number entering a hut without a bednet), and on k_p (the proportion which enter and successfully feed a p-treated hut), given that the latter already includes entering.*

We thank the reviewer for highlighting this section as a mistake was made when transcribing the parameter definitions from the original Griffin et al. model. k_p is actually the probability of successful feeding conditional on the mosquito having entered, not the proportion which enter and successfully feed as we initially stated. We have corrected the definition in the main text thus:

“The proportion of mosquitoes entering the hut which exit without feeding is denoted j_p whilst the proportion which successfully feed upon entering is k_p.”

*Moreover, it is claimed that s_p0 is a proportion, so should be constrained between 0 and 1, but if m_p = 1 (so bednets have no effect on entering) then s_p0=k_p/k_0, which is a ratio of two proportions rather than a proportion itself (for plausible parameters values it presumably will be below 1, but it is not necessarily so).*

We agree that this has not been properly explained. We have tightened up the definition of s_p0 in the main text:

“Estimates of j_p, l_p and m_p can be used to determine the proportion of mosquitoes repeating (a combination of deterrence and exiting, r_p0), dying (d_p0) and feeding successfully (s_p0) during a single feeding attempt in a hut with a new LLIN relative to those successfully feeding in a hut without an LLIN (i.e. p=1 or 2)…”.

And given a further justification for the assumptions implicit in the equations in the following paragraph (using data from the meta-analysis):

“Though theoretically s_p0 could have values >1 for practical purposes it is constrained between zero and one as on average mosquitoes entering a hut with an LLIN are less likely to feed than mosquitoes entering a hut without a bednet (as shown by all estimates of k_p being substantially lower than k_0, see Figure 3 and Table 6).”

*I therefore think further motivation is required, which should ideally include some graphical assessments of model assumptions (3rd order polynomial, normally distributed errors, relationship between k_p and mortality) which could go in the appendix.*

*It is also good practice in any Bayesian analysis to explicitly show posterior distributions of model parameters or at the very least to summarise CrIs for these.*

We agree that further graphical assessment would allow the reader to have a better understanding of the validity of the model. Therefore we have included all the figures highlighted by the reviewer. A plot justifying the use of a normal distribution when fitting m_p is included in Figure 3—figure supplement 1 whilst posterior distributions of each of the 14 other parameters is shown in Figure 3—figure supplement 1). The relationship between k_p and mortality (in experimental hut trials) is shown in Figure 3 whilst the relationship between s_p0 and mortality (in the bioassay) is shown in Figure 4—figure supplement 1). All figures have been referred to in the main text or figure captions.

*Reviewer #1 (Additional data files and statistical comments):*

*It would be useful to provide code and data used in meta analyses – and the statistical submission form states that these will be made available.*

The code is included in the resubmission, together with the initial dataset that will be added to Dryad.

*Reviewer #2:*

*The authors have used three small datasets (all that is available) and many assumptions to model the impact of insecticide resistance, defined using insecticide bioassay results, on malaria incidence in Africa in a defined range of scenarios. Their scope is ambitious and this work brings together several separate studies to incorporate resistance into an existing malaria model. The work builds on earlier studies and is novel and interesting, however, the results need to be caveated carefully to recognise the limitations of the datasets used and the assumptions made. It also needs to be clear that this work makes predictions for a limited range of scenarios.*

*The work uses a series of meta-analyses to show that insecticide bioassay results can be used to predict the impact of resistance on malaria incidence, however, the final model has not been validated using African locations that were not included in the meta-analyses datasets where values are known for both malaria incidence and resistance as measured by an insecticide bioassay.*

The reviewer raises an important point and we agree that there should be further statements for the limitation of the model given the available datasets. Specific changes are outlined below. Regarding the validation of the model with field data we would very much like to do this exercise though are unaware of any current publicly available datasets which would allow such validation. To our knowledge there are no sites where insecticide bioassays are regularly measured together with malaria metrics (either prevalence or incidence) and where the history of malaria control is known. If the reviewer knows of such a dataset we would be happy to try and test out the model. Hopefully studies will be forthcoming over the next few years which would allow for elements of the model to be validated in specific locations. We have modified the following paragraph to raise the important question of model validation:

“Currently a mosquito population is defined as being pyrethroid resistant if there is <90% bioassay mortality (World Health Organisation, 2013; Viana et al., 2016). […] Though the principles outlined here may apply to other mosquito species in different settings care should be taken when extrapolating the results beyond the areas where the data were collated.”

*The data used, the mosquito species included and the range of slide prevalence values considered are all specific to scenarios found in Africa but nowhere in the paper is this limitation mentioned. The implication from the title and throughout the manuscript is that this is a generalizable analysis of pyrethroid resistance but it is in fact only applicable in Africa, and only in certain scenarios within subsaharan Africa.*

We agree that the work is very African centric so have changed the title of the paper to reflect this (see below). Further clarification of the geographical and species limits of the model predictions is outlined in our response above to Reviewer #1.

Throughout the Results and Discussion there are sentences that look like statements of fact but they are in fact predictions within the limits of certain scenarios defined by the authors and predicated on analysis of a limited dataset and many assumptions, for example, "For the An. gambiae complex PBO had the greatest benefit in mosquito populations with intermediate levels of pyrethroid resistance", "The probability that a mosquito will feed on someone beneath a LLIN only increases at high levels of pyrethroid resistance", and so on. These statements need to be presented as predictions, and it would also be useful if the Results could start with a summary of the scenarios covered and assumptions made (see below).

We have gone through the Results and Discussion sections and clarified what evidence we are using with all of our statements. Broadly this breaks down into data observations and model predictions. Each change has been noted below (examples where single words were switched are not included).

Results – “Data suggests that for the *An. gambiae* complex PBO has the greatest benefit in mosquito populations with intermediate levels of pyrethroid resistance”;

Results – “Figure 3 indicates that the number of mosquitoes deterred from entering the experimental hut substantially decreases in areas of higher pyrethroid resistance…”;

Results – “The transmission dynamics model predicts that the higher the level of pyrethroid resistance the greater impact it will have on both the number of clinical cases (Figure 4)”;

Results – “The impact of the addition of the synergist, PBO, on pyrethroid induced mortality appears to depend on mosquito species and the level of pyrethroid resistance”;

Discussion – “The meta-analysis of experimental hut trial data suggest that the probability that a mosquito will feed on someone beneath a LLIN only increases at high levels of pyrethroid resistance (Figure 3)”;

Discussion – “The loss of LLIN induced mosquito mortality *is likely* to decrease the community impact of LLINs, increasing average mosquito age and the likelihood that people are infected whilst unprotected by a bednet.”

*In Results, the authors state "The numbers of mosquitoes deterred from entering the experimental hut substantially decreases in areas of higher pyrethroid resistance" – this is a strong statement but when you look at Figure 3 you can see that the credible intervals are large ("substantial" even) and this statement needs to caveated appropriately.*

We agree and have changed the text accordingly:

“Figure 3 indicates that the number of mosquitoes deterred from entering the experimental hut substantially decreases in areas of higher pyrethroid resistance (where LLIN induced mortality inside the hut is low) though the variability around the best fit line is high suggesting the precise shape of the relationship is uncertain.”

*Where values are given, these are not bounded by any intervals, for example in Results "causing up to 200 additional cases per 1000" or "where over 500 cases per 100 people can be prevented each year". A range or intervals are needed to give an idea of uncertainty.*

We agree that credible intervals would help interpretation and have included them were necessary together with full credible intervals estimates for Figure 4—figure supplement 1-3 and Figure 5—figure supplement 1–Figure 5—figure supplement 3.

“For example with as little as 30% resistance (70% mortality in discriminating dose assay) in a population with 10% slide prevalence (in 2-10 year olds) the model predicts that pyrethroid resistance would cause an additional 245 (95%CI 142-340) cases per 1000 people per year (Figure 4, averaged over the 3 year life-expectancy of the net)”.

“For example in an area with 10% endemicity and 80% resistance (20% mortality in discriminating dose assay) the model predicts that switching to PBO LLINs would avert 501 (95%CI 319-621) cases per 1000 people per year (Figure 5).”

*Meta-analyses 1 and 2 were compared to the observed results collected for the third meta-analysis but this comparison was visual only with no formal analysis or validation (end of subsections “Added benefit of PBO” and” Predicting the added benefit of PBO LLINs in experimental hut trials”).*

We understand that some readers might like to see some sort of test to show the goodness of fit. This analysis has been done and highlighted in the Methods and caption of Figure 2:

“Overall the model appears to be a good predictor of these data, both visually and statistically (Analysis of Variance test shows there was no significant difference between model predictions and observed data p-value=0.35).”

*The paper as a whole, and the Methods in particular, is a long and dense read and would benefit from summaries for biomedical/entomological readers who are not mathematical modellers, whilst still retaining important details about the scenarios modelled and the assumptions made. In particular, summaries aimed at these readers at the beginning of both the Methods and Results would be hugely helpful for eLife's broader readership.*

We agree that the paper is dense, though given the depth required to explain the diverse range of methods we think it is essential. As suggested we have tried to summarise the Methods section by extending the outline of the paper at the end of the Introduction (see below or the extended paragraph). As for a summary of the results we feel that this will be best done as part of the *eLife* digest, which will be written if the paper is accepted.

“Here we propose that information on the current malaria endemicity, mosquito species and level of pyrethroid resistance (as measured by bioassay mortality) can be used to predict the public health impact of pyrethroid resistance and choosing the most appropriate LLIN for the epidemiological setting. […] Finally (4) this model is combined with bioassay and experimental hut trial results to predict the epidemiological impact of switching from mass distribution of a standard to a PBO LLIN.”

*Figure 1 is cited in the Introduction and seems to be a key result but is not explained in the Results at all or discussed. No credible intervals are included. It shows model predictions for the scenario where bioassay results give 20% mortality but it would be interesting to see the results for other mortality/survival rates that are often found in the region.*

Figure 1 is intended to be used to interpret the results of the transmission dynamics model. It illustrates the scenarios and introduces the variables which are changed in the later figures. Therefore it originally wasn’t highlighted in the Results, just the Introduction and Methods. Having said this we agree with the reviewer that showing reruns of the model with different parameters might be informative. Therefore we have included two additional figures supplements to Figure 1 with two different levels of pyrethroid resistance, low (20% bioassay survival, Figure 1—figure supplement 1) and high (80% bioassay mortality, Figure 1—figure supplement 2). We now present the moderate scenario (50% bioassay mortality) in Figure 1. The new figures are in addition referred to in the main Discussion.

“Detecting an epidemiological impact of a low population prevalence of resistance may be challenging for local health systems (for example, see <20% resistance prevalence Figure 1—figure supplement 1, Figure 4) especially in an area where LLIN coverage, local climatic conditions and the use of other malaria control interventions are changing over time. These simulations also assume that resistance arrives overnight, when in reality it will spread through a mosquito population more gradually and therefore may be harder to detect.”

*The green dashed line in 1A shows 10% parasite prevalence but it is unclear why. The starting prevalence is >20% and then it drops after control but come up to 10% every three years so presumably this is the setting the green line refers to?*

Yes, it is. It is mentioned in the caption for the figure that “Endemicity (a variable in Figure 4 and Figure 5) is changed by varying the slide prevalence in 2-10 year olds at year 6 (by changing the vector to host ratio) and in this plot takes a value of 10% (as illustrated by the horizontal green dashed line in A).” To stress the importance of Figure 1 to the interpretation of later Figures we have added the following section to the end of the Introduction:

“Thirdly, information from (1) and (2) is used to parameterise a widely used malaria transmission dynamics mathematical model to estimate public health impact of pyrethroid resistance in different settings taking into account the community impact of LLINs. An illustration of model predictions showing how different malaria metrics change over time is given in Figure 1. The figure also indicates how LLIN coverage and variables such as malaria endemicity and pyrethroid resistance are incorporated in the model.”

*The legend says the black line shows the situation with no resistance and the red line shows the situation if resistance arrives at Y6, but the red lines starts before Y0 and the black line doesn't start until Y6.*

In the original figure the red line (resistance population) overran the black line (susceptible population) until Y6 when they diverge. To avoid confusion we have added a little jitter so that both lines are visible throughout.

*There are a lot of assumptions made by this work but it is unclear how they have been justified and which ones have been subject to sensitivity analyses in the context of the results presented here, i.e. the impact of the prevalence of resistance on malaria incidence.*

We agree that there are lots of assumptions which are generally made due to lack of available data. When this is the case we have resorted to the simplest explanation first though each have been expanded upon in the text in turn.

“Visual inspection of these data indicates that mosquito deterrence and exiting can be described by the degree of mosquito mortality seen in the same hut trial.”

“As there is insufficient information to generate these functions for each species separately it is assumed the relationship between deterrence/exiting, feeding successfully and dying is consistent across all species”.

“For simplicity and following (Griffin et al.) it is assumed that the killing activity of pyrethroid over time is proportional to the loss of morbidity caused by washing…”.

“Following Griffin et al. itis assumed that the activity of the insecticide decays at a constant rate according to…”.

*The assumptions made include: Assumed mosquito deterrence and exiting can be described by the degree of mosquito mortality seen in the same hut trial; Assumed the relationship between deterrence/exiting, feeding successfully and dying is consistent across all species; Assumed washing nets gives the same results as a durability study; Assumed the activity of the insecticide decays according to a given formula that includes half-life in washes; Assumed resistance arises spontaneously, and after six years of LLINs use; Assumed LLINs are re-distributed every three years; Assumed transmission is perennial; Assumed there is no other vector control (and presumably no other non-vector related pyrethroid use); Assumed resistance remains constant after arising; Assumed 35% clinical cases are treated of which 36% receive ACT);*

Each of the above assumptions are made to produce a generalisable site in Africa as in reality every geographical location will have different seasonality, level of treatment and vector control interventions and pyrethroid resistance profile. It is infeasible to do a sensitivity analysis on all these assumptions and unlikely to provide the reader with an additional insight. Therefore throughout the manuscript we only vary the 3 most informative settings, mosquito species, malaria endemicity and level of pyrethroid resistance (as measured by a bioassay).

*Incorporated assumptions/estimates used previously by the same group that are not all given here;*

The number of individual assumptions and parameter estimates made in the Griffin et al. and Walker et al. papers are considerable and including them all here would make this manuscript unwieldly. Therefore we refer the reader to the original references.

*Assumed physiological resistance has no effect on the vectorial capacity of individual mosquitoes.*

There are a multitude of ways in which resistance might impact vectorial capacity other than changing the susceptibility of mosquitoes to insecticide. However, to our knowledge none of these have been shown to occur in wild caught mosquitoes so including them at this stage would be premature and is beyond the scope of this paper. We have stressed this in the following sentence at the start of the Results:

“It is assumed that the ability of a mosquito to survive insecticide exposure is not associated with any other behavioural or physiological change in the mosquito population which influences malaria transmission. For example, an increased propensity for mosquitoes to feed outdoors (subsequently referred to as behavioural resistance) would limit their exposure to LLINs though there is currently insufficient field evidence to justify its inclusion in the model (Briet and Chitnis, 2013; Gatton et al., 2013).”

*Reviewer #2 (Additional data files and statistical comments):*

*I am not a mathematical modeller and assume that this manuscript has also gone to reviewers who can comment on the modelling work in more depth.*

*The authors propose to make the data used in the meta-analysis available via Dryad if they have previously been published. This is reasonable but some of the unpublished datasets were provided by the authors of this paper and so should also be included in data deposit.*

We agree that ideally all data would be made immediately publicly available on acceptance of this manuscript. All bioassay data (published and unpublished) shall be uploaded to Dryad though some experimental hut trial data are currently unavailable for inclusion. We contacted all custodians of the unpublished experimental hut trial data and a number of them would prefer for their raw data not to be made publicly available as they are currently in the process of publishing it themselves. Once these manuscripts have been published we will update the Dryad dataset so that it includes all data points reviewed in this study.